# RHEB/mTOR hyperactivity causes cortical malformations and epileptic seizures through increased axonal connectivity

**Martina Proietti Onori**[1,2], **Linda M. C. Koene**[1,2☉], **Carmen B. Schäfer**[1☉], **Mark Nellist**[3], **Marcel de Brito van Velze**[1], **Zhenyu Gao**[1], **Ype Elgersma**[1,2,3]*, **Geeske M. van Woerden**[1,2,3]*

**1** Department of Neuroscience, Erasmus Medical Center, Rotterdam, the Netherlands, **2** The ENCORE Expertise Center for Neurodevelopmental Disorders, Erasmus Medical Center, Rotterdam, the Netherlands, **3** Department of Clinical Genetics, Erasmus Medical Center, Rotterdam, Zuid Holland, the Netherlands

☉ These authors contributed equally to this work.
* y.elgersma@erasmusmc.nl (YE); g.vanwoerden@erasmumc.nl (GMvW)

**Data Availability Statement:** All relevant data are within the paper and its Supporting Information files.

## Abstract

Hyperactivation of the mammalian target of rapamycin (mTOR) pathway can cause malformation of cortical development (MCD) with associated epilepsy and intellectual disability (ID) through a yet unknown mechanism. Here, we made use of the recently identified dominant-active mutation in *Ras Homolog Enriched in Brain 1* (*RHEB*), RHEBp.P37L, to gain insight in the mechanism underlying the epilepsy caused by hyperactivation of the mTOR pathway. Focal expression of RHEBp.P37L in mouse somatosensory cortex (SScx) results in an MCD-like phenotype, with increased mTOR signaling, ectopic localization of neurons, and reliable generalized seizures. We show that in this model, the mTOR-dependent seizures are caused by enhanced axonal connectivity, causing hyperexcitability of distally connected neurons. Indeed, blocking axonal vesicle release from the RHEBp.P37L neurons alone completely stopped the seizures and normalized the hyperexcitability of the distally connected neurons. These results provide new evidence of the extent of anatomical and physiological abnormalities caused by mTOR hyperactivity, beyond local malformations, which can lead to generalized epilepsy.

## Introduction

Malformations of cortical development (MCD) are a heterogenous group of micro- and macroscopic cortical abnormalities, such as focal cortical dysplasia (FCD), megalencephaly, lissencephaly, and periventricular nodular heterotopia [1]. MCD arise from disturbances in cortical development during early embryogenesis and are often linked to epilepsy and intellectual disability (ID) [2–4]. It is estimated that up to 40% of intractable or difficult to control childhood seizures are due to MCD, and vice versa, at least 75% of the patients with MCD will develop seizures [5].

The mammalian target of rapamycin (mTOR) is a kinase that mediates many cellular processes, including neuronal progenitors proliferation and cell growth [6,7]. mTOR forms 2 distinct protein complexes, mTORC1 and mTORC2, each characterized by different binding

**Funding:** This work was supported by the Dutch TSC foundation (STSN) (G.M.v.W) and the Dutch Epilepsy foundation (Epilepsie fonds) (Y.E.). GMvW was supported by the Dutch Research Counsil (NWO) Vidi Grant (016.Vidi.188.014). The funders had no role in study design, data collection and analysis, decision to publish, or preparation of the manuscript.

**Competing interests:** The authors have declared that no competing interests exist.

**Abbreviations:** ABC, avidin-biotinperoxidase complex; ACSF, artificial cerebrospinal fluid; AED, antiepileptic drug; BDNF, brain-derived neurotrophic factor; DAPI, 4′,6-diamidino-2-phenylindole; DIV1, 1 day in vitro; DMEM, Dulbecco's modified Eagle medium; EEG, electroencephalography; EPSC, excitatory postsynaptic potential; FCD, focal cortical dysplasia; GAP, GTPase activating protein; HFO, high-frequency oscillation; ID, intellectual disability; IUE, in utero electroporation; L2/3, layer 2/3; LFP, local field potential; LSL, Lox-Stop-Lox; MCD, malformation of cortical development; mTOR, mammalian target of rapamycin; NHS, normal horse serum; PB, phosphate buffer; PFA, paraformaldehyde; *RHEB, Ras Homolog Enriched in Brain 1*; RHEB WT, wild-type RHEB; SScx, somatosensory cortex; TeTxLC, tetanus toxin light chain; TSC, tuberous sclerosis complex; TTX, tetrodotoxin.

partners [8]. Ras Homolog Enriched in Brain 1 (RHEB), a member of the RAS family of small GTPases, is the direct activator of mTORC1 [9,10]. The conversion of active GTP-bound RHEB to the inactive GDP-bound form is mediated by the tuberous sclerosis complex (TSC), which acts as a RHEB GTPase activating protein (GAP) [9,11]. In response to nutrients and growth factors, the TSC complex is inhibited, allowing activation of mTORC1 by RHEB-GTP [12,13]. Studies in *Rheb* knock-out mice showed that RHEB activity is the rate-limiting step for mTOR activation in the brain and that neuronal functioning in particular is sensitive to increased RHEB-mTOR signaling [14].

Hyperactivation of the mTOR pathway by mutations in genes encoding components of the mTOR pathway (e.g., *AKT3, PIK3CA, DEPDC5, PTEN, TSC1, TSC2, RHEB*, and *MTOR* itself) has been associated with different types of MCD, such as megalencephaly and FCD, as well as with epilepsy [2,15,16]. The underlying genetic variability explains the heterogeneity of MCD and illustrates the challenges involved in understanding the mechanisms underlying MCD-associated epilepsy.

The discovery of genetic mutations that cause FCD or other types of MCD allowed the generation of animal models to study the development of MCD and associated epilepsy [17,18]. In particular, in utero electroporation (IUE), which allows for the spatial and temporal control of transgene expression during embryonic development, has been used to generate mouse models with focal malformations and epilepsy [19–21].

One recent FCD mouse model was generated by using IUE to overexpress the constitutively active RHEBp.S16H mutant [22]. This results in mTOR hyperactivity, FCD, and spontaneous seizures [20]. Recently, we identified 2 de novo mutations in *RHEB* (c.110C>T (p.P37L) and c.202T>C (p.S68P)) in patients with ID, epilepsy, and megalencephaly [23], providing for the first time a clinically relevant link between RHEB and MCD. IUE of a construct encoding the RHEBp.P37L mutant caused severe focal cortical lesions, resembling periventricular nodular heterotopia, and diffuse neuronal misplacement in the cortex. Furthermore, the mice reliably developed spontaneous seizures starting at 3 weeks of age [23]. The anatomical and phenotypical features of this novel mouse model, fully recapitulating the most prominent characteristics of MCD (focal lesions and epilepsy), make this a powerful tool and clinically relevant novel model to study the mechanisms underlying mTOR and MCD-related epilepsy.

Using the patient-related RHEBp.P37L mutation, we provide evidence that persistent activation of the mTOR pathway results in anatomical and functional changes in axonal connectivity and that this causes increased excitability of distally connected neurons and the development of generalized seizures.

## Results

### The RHEBp.P37L protein is resistant to TSC complex inhibition and causes aberrant cortical development in vivo

The RHEBp.P37L mutation was identified in patients with ID, megalencephaly, and epilepsy, and it was proposed to act as a gain of function mutation [23], although it was not clear how. One possibility is that the mutation renders RHEB resistant to the GAP function of the TSC complex. To assess whether the TSC complex can convert RHEBp.P37L from its active GTP- to its inactive GDP-bound state, we compared the effects of transient in vitro overexpression of the RHEBp.P37L mutant with wild-type RHEB (RHEB WT) and the RHEBp.S16H mutant, a well-known gain-of-function mutant of RHEB recently used to generate an FCD mouse model [20,22]. In the absence of TSC, overexpression of RHEB WT as well as both RHEB mutants caused increased mTORC1 activity, as measured by T389-phosphorylation of coexpressed S6K, a direct substrate of the mTORC1 kinase (**Fig 1A**, see **S1 Table** for statistics

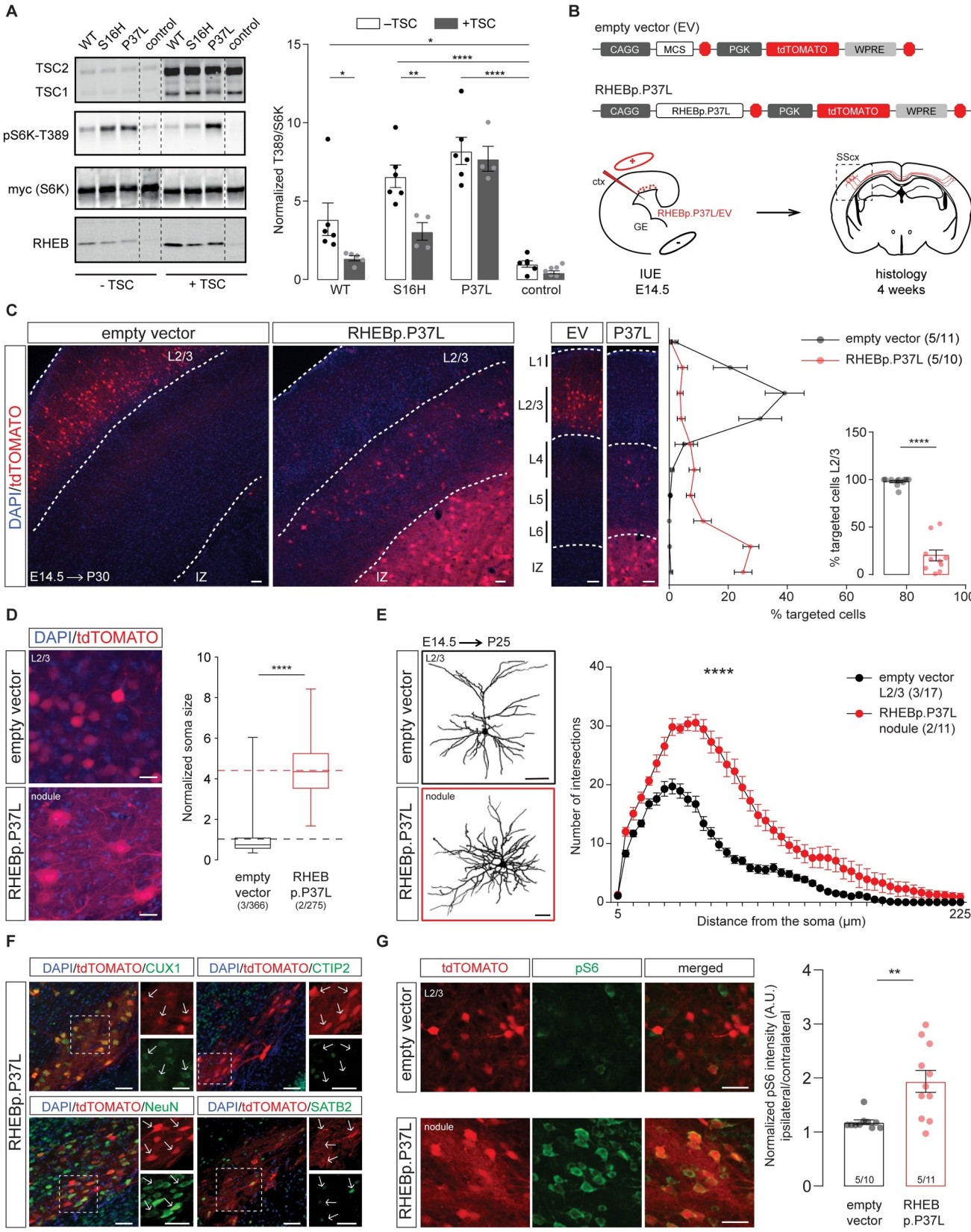

**Fig 1. The RHEBp.P37L protein is resistant to TSC complex inhibition and causes aberrant cortical development in vivo. (A)** RHEB WT, p.S16H, or p.P37L constructs were transiently coexpressed with an S6K reporter construct and the TSC complex in HEK 293T cells to assess the effect on mTORC1 activity. Quantification of the ratio of T389-phosphorylated S6K to total S6K was calculated relative to control condition, in absence or presence of the TSC complex (control indicates empty vector pcDNA3); dashed lines indicate where an irrelevant lane in the original scan was excluded from the picture; bar graph represents mean ± SEM and single data points represent the number of independent biological samples per condition; for statistics, see S1 Table. **(B)** Schematic representation of the main constructs used throughout the experiments and overview of the experimental design. MCS indicate the multiple cloning site with specific restriction sites (AscI and PacI in this case) to insert the gene of interest. Each construct was delivered by IUE at E14.5 to target the progenitor cells of L2/3 pyramidal neurons in the SScx. **(C)** Representative confocal images of the targeted SScx counterstained with DAPI showing the transfected cells (tdTomato+) in red (see also S1A Fig) and quantification of tdTomato+ cells across the different layers of the SScx with percentage of cells reaching L2/3 in the inset (bins 1–5 from the top). Dotted lines indicate the border of the IZ (bottom) and delineate L2/3. Numbers in the legend indicate number of targeted mice ($N = 5$) and total number of pictures analyzed ($n = 11$, $n = 10$); results are represented as mean ± SEM and single data points in the bar graph indicate the number of pictures analyzed; inset analysis: Mann–Whitney $U = 0$, $p < 0.0001$, two-tailed Mann–Whitney test. **(D)** Soma size quantification of L2/3 empty vector expressing cells and RHEBp.P37L expressing cells in the nodule; box plots represent minimum and maximum values with median, dashed lines represent the mean values for empty vector (black) and for RHEBp.P37L (red); numbers indicate number of targeted mice ($N = 2$, $N = 3$) and number of cells analyzed ($n = 275$, $n = 366$); Mann–Whitney $U = 1940$, $p < 0.0001$, two-tailed Mann–Whitney test. **(E)** Reconstruction and Sholl analysis of dendritic morphology of biocytin filled cells in L2/3 of the SScx (for empty vector control) and RHEBp.P37L cells in the nodule; numbers in the legend indicate number of targeted mice ($N = 3$, $N = 2$) and number of cells analyzed ($n = 17$, $n = 11$); data are presented as mean ± SEM; interaction group condition/distance from the soma: $F_{(44, 1144)} = 15.69$, mixed-effects analysis; $p < 0.0001$. **(F)** Representative images of the nodule stained with CUX1 (L2/3 marker), CTIP2 (L5 marker), SATB2 (cortical projection neurons marker), or NeuN (mature neurons marker); arrows in the zoomed pictures point at examples of targeted cells (for an overview, see S1B Fig). **(G)** Representative images of the targeted L2/3 (SScx) of empty vector control and nodule showing increased pS6-240 levels for the ipsilateral targeted cortex in RHEBp.P37L targeted mice; for an overview, see S1D Fig; bar graph represents mean ± SEM and single data points indicate the values of each normalized ipsilateral/contralateral pS6 intensity; numbers in the bars indicate number of targeted mice ($N = 5$) and number of pictures analyzed ($n = 10$, $n = 11$); Mann–Whitney $U = 13$, $p = 0.002$, two-tailed Mann–Whitney test. Histological analysis for **(D), (F),** and **(G)** was performed on 5-week-old mice. The data underlying this figure can be found in S1 Data. $^{*}p < 0.05$, $^{**}p < 0.01$, $^{****}p < 0.0001$; scale bars: 20 μm (D), 50 μm (C, E–G). ctx, cortex; DAPI, 4′,6-diamidino-2-phenylindole; EV, empty vector; GE, ganglion eminence; IUE, in utero electroporation; IZ, intermediate zone; L2/3, layer 2/3; MCS, multiple cloning site; RHEB, Ras Homolog Enriched in Brain 1; RHEB WT, wild-type RHEB; SScx, somatosensory cortex; TSC, tuberous sclerosis complex.

overview). In the presence of the TSC complex, mTORC1 activity was reduced in the RHEB WT and RHEBp.S16H expressing cells but not in the RHEBp.P37L expressing cells (Fig 1A). Here, presence or absence of the TSC complex resulted in similar levels of S6K phosphorylation, confirming that the patient-derived RHEBp.P37L acts as a gain of function mutation, which is resistant to the inhibitory action of the TSC complex (Fig 1A).

Using IUE, we have previously shown that overexpression of the RHEBp.P37L mutant, but not RHEB WT, results in the formation of a heterotopic nodule as well as spontaneous epilepsy in 100% of the targeted mice [23], providing us with a valuable model to study the mechanisms behind mTORC1-dependent and MCD-related epilepsy. To confirm previous results, we used IUE to introduce the RHEBp.P37L expression vector or an empty vector control at E14.5 in progenitor cells that give rise to layer 2/3 (L2/3) pyramidal neurons of the somatosensory cortex (SScx) (Fig 1B). As shown previously, overexpression of RHEBp.P37L resulted in a clear migration deficit, with only 20% of the targeted cells reaching the outer layers of the cortex (L2/3) compared to 97% in the empty vector condition (Fig 1C, inset). The nonmigrated transfected neurons remained in the white matter to form a heterotopic nodule lining the ventricle in the adult brain (Fig 1C and S1A Fig).

Further characterization of the RHEBp.P37L-dependent MCD revealed that while the general cortical layer architecture remained intact (S1A Fig), ectopic RHEBp.P37L overexpressing cells showed cytological abnormalities, with dysmorphic appearance and enlarged soma size (Fig 1D and S1A Fig). Sholl analysis of biocytin filled cells in the heterotopic nodule of RHEBp.P37L expressing neurons revealed that the cells in the nodule presented a more complex arborization compared to empty vector control cells in L2/3 (Fig 1E). Transfected ectopic neurons preserved the molecular identity of mature L2/3 pyramidal cells, being positive for the neuronal marker NeuN and the outer layer molecular marker CUX1 and negative for the deeper layer marker CTIP2 (Fig 1F and S1B Fig). Additionally, 80% of the targeted neurons were SATB2 positive, showing that, despite being mislocalized, they maintained the excitatory callosal projection identity (Fig 1F and S1B Fig). Importantly, RHEBp.P37L-transfected cells

were negative for interneuron-specific markers such as GABA and PV (**S1C Fig**). Finally, mice overexpressing RHEBp.P37L showed an overall increase in ribosomal protein S6 phosphorylation, a commonly used readout for mTORC1 activity, in the transfected (ipsilateral) SScx compared to the empty vector condition (**Fig 1G** and **S1D Fig**).

## Overexpression of RHEBp.P37L in vivo causes mTORC1-dependent spontaneous generalized tonic–clonic seizures and abnormal neuronal network activity

In order to study the mechanism behind mTOR-dependent MCD-related epilepsy, we next assessed the reliability of seizure development using our RHEBp.P37L mutant model. To assess the reliability of spontaneous seizure development, the RHEBp.P37L mice were continuously monitored from weaning (P21) until at least 2 months of age, using wireless electroencephalography (EEG) with electrodes placed bilaterally on the SScx (**Fig 2A**). Spontaneous seizures started to appear in all RHEBp.P37L transfected mice ($N$ = 12), at 3 weeks of age, with an average onset of 33 days, while none of the control mice ($N$ = 6) developed any epileptic events, confirming previous data [23] (**Fig 2B** and **S2A Fig**). These seizures were highly stereotypical, characterized by the loss of upright posture followed by a tonic–clonic phase with convulsions and twitching behavior. EEG analysis showed that the seizures were characterized by an increase in frequency and amplitude of brain activity (**Fig 2C**, box 3 ictal activity) compared to baseline interictal activity (**Fig 2C**, box 2) and baseline activity of control mice (**Fig 2C**, box 1). The calculated average duration of an epileptic event was 40 seconds (mean ± SEM: 42.6 ± 1.33), followed by a post-ictal depression phase of variable length (**Fig 2C**, box 4 post-ictal activity). The average number of seizures per day, measured across multiple EEG sessions over at least 2 consecutive days for each session, was 4 (mean ± SEM: 3.8 ± 0.76), with variability between mice as well as over time (**S2B Fig**). Additionally, no correlation was found between the average number of seizures over all the recording days and the average number of targeted cells per mouse, confirming previous literature [24] (**S2C Fig**).

Electrographic frequency dynamics of the interictal phases, and especially *theta* oscillations, have proven to be good predictors for epilepsy outcome, compared to epileptiform spikes or high-frequency oscillations (HFOs), in several rodent models of epilepsy [25,26]. Therefore, using local field potential (LFP) recordings, with the recording electrodes placed bilaterally in the SScx at the depth of L2/3, we assessed the frequency dynamics of cortical brain activity in the interictal periods of RHEBp.P37L expressing mice, starting from 4 weeks of age (**Fig 2D**). The normalized averaged power spectrum of the RHEBp.P37L group did not reveal a significant difference between the ipsilateral and contralateral (nontargeted) SScx (targeting: F(1, 22 = 1.43, $p$ = 0.25, nonsignificant, two-way repeated measure ANOVA); therefore, measurements from both sides were pooled. Whereas the total power across 5 days of recording did not differ between the RHEBp.P37L ($N$ = 12) and the control group ($N$ = 8) (Mann–Whitney $U$ = 157, $p$ = 0.35, nonsignificant, two-tailed Mann–Whitney test), a significant difference in the *delta* (2 to 4 Hz), *theta* (4 to 8 Hz), and *gamma* (30 to 50 Hz) frequency bands of the normalized power spectrum was seen in the RHEBp.P37L group compared to the control group (**Fig 2E and 2F** and **S2D Fig**; statistics in **S2 Table**). The difference in the *theta* and *gamma* frequency bands, but not in the *delta*, could be reverted to the control condition by injecting the RHEBp.P37L mice with 10 mg/kg rapamycin intraperitoneally for 3 consecutive days (**Fig 2E and 2F** and **S2D and S2E Fig**; statistics in **S2 Table**), proving that in our model, the seizures are caused by hyperactivity of the mTOR pathway. Additionally, this result indicates that *theta* oscillations, which negatively correlate with *gamma* frequencies [26], are a good predictor for epileptogenesis in hyperactive mTOR-dependent models.

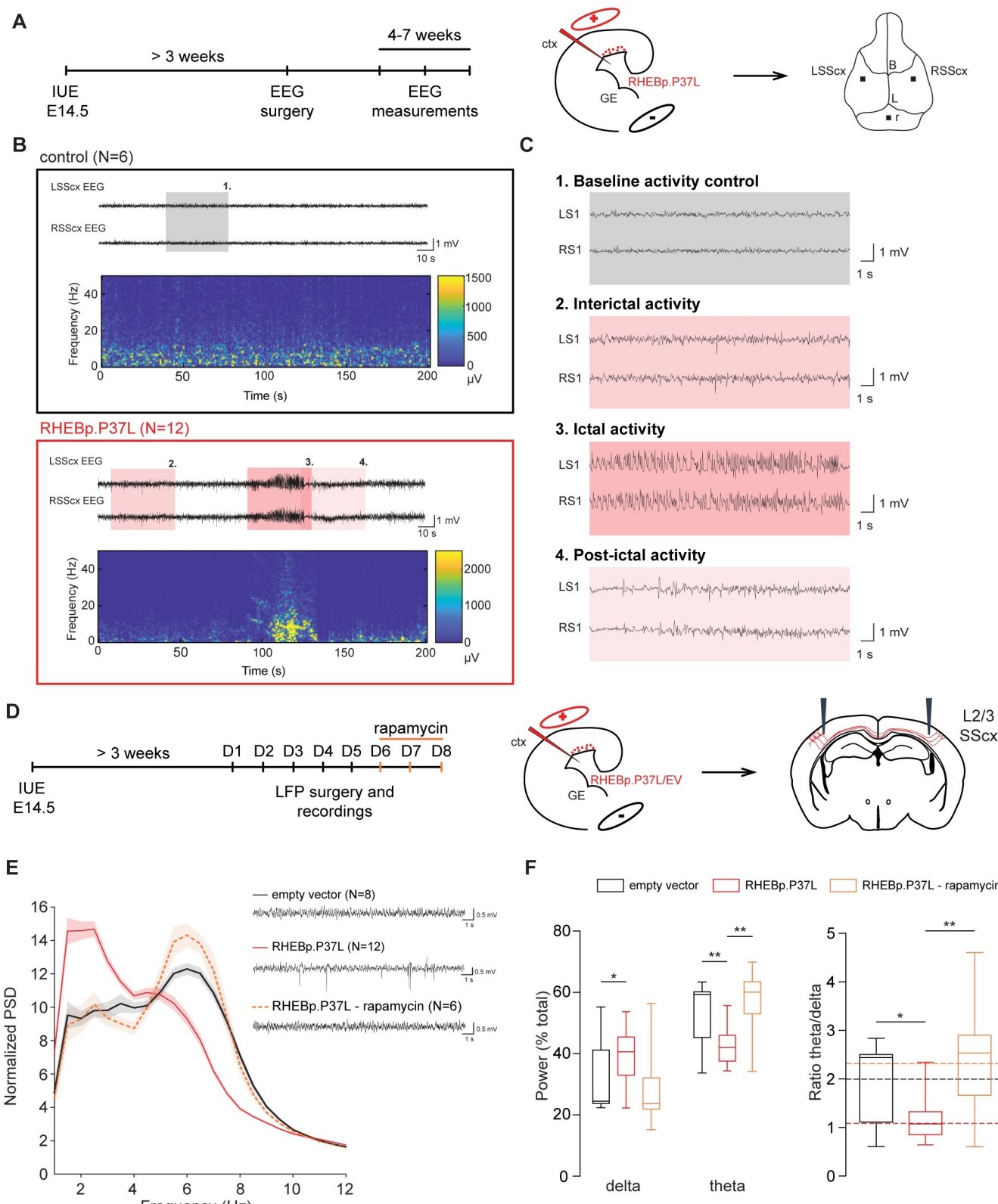

**Fig 2. Overexpression of RHEBp.P37L in vivo causes mTORC1-dependent spontaneous generalized tonic–clonic seizures and abnormal neuronal network activity. (A)** Timeline and experimental design indicating the cortical area targeted with the IUE and position of the electrodes placed during the

EEG surgery. **(B)** Example EEG traces and spectrogram of 5-week-old control mouse ($N = 6$, nontargeted mice from the same litters as the RHEBp.P37L mice) and RHEBp.P37L mouse ($N = 12$); see also **S2A–S2C Fig**; colored boxes are zoomed in panel **(C)**. **(C)** Highlighted EEG traces showing: box 1, the baseline activity of a control mouse; box 2, the interictal activity; box 3, the ictal (seizure) activity; and box 4, the post-ictal phase of a RHEBp.P37L targeted mouse. **(D)** Timeline and experimental design indicating the cortical area targeted with the IUE, the position of the electrodes for the LFP recordings, and the IP rapamycin injections. **(E)** Example LFP traces for each group condition and normalized PSD averaged bilaterally over the overall consecutive days of recording (for the PSD until 50 Hz, see **S2D Fig**); N indicates number of mice analyzed for each group; data are represented as mean (thick line) ± SEM (shading area). **(F)** Calculation of the *delta* (2–4 Hz) and *theta* (4–8 Hz) frequency bands over the total power of the PSD presented in **(E)**, and relative ratio *theta/delta* (see also **S2E Fig**); box plots represent minimum and maximum values with median, dashed lines represent the mean values for each group; for statistics, see **S2 Table**; the data underlying this figure can be found in **S2 Data**. *$p < 0.05$, **$p < 0.01$, ***$p < 0.001$. B, bregma; ctx, cortex; EEG, electroencephalography; EV, empty vector; GE, ganglion eminence; IP, intraperitoneal; IUE, in utero electroporation; L, lambda; L2/3, layer 2/3; LFP, local field potential; LSScx, left SScx; PSD, power spectrum density; r, reference electrode; RSScx, right SScx; SScx, somatosensory cortex.

## The heterotopic nodule is neither necessary nor sufficient to induce spontaneous seizures

Cortical malformations occur during early embryonic development and are generally associated with the development of epileptic activity [4]. Therefore, transient treatment with mTOR inhibitors during brain development might prevent the formation of a cortical malformation and could consequently reduce the chances of developing epilepsy. To assess if early transient down-regulation of the mTORC1 pathway upon overexpression of RHEBp.P37L could prevent the development of heterotopic nodules, we injected pregnant female mice with 1 mg/kg of rapamycin for 2 consecutive days starting 1 day after IUE of the RHEBp.P37L vector (**Fig 3A**). Prenatal down-regulation of the mTORC1 pathway significantly improved the migration of the targeted neurons, with 75% of the targeted cells successfully migrating out (**Fig 3B**). In addition, prenatal rapamycin treatment prevented the formation of a heterotopic nodule in 9 out of 11 mice. However, 7 out of the 11 targeted mice (58%) still showed spontaneous seizures, including 5 mice that did not develop a discernable heterotopic nodule, with an average number of seizures per day similar to RHEBp.P37L mice not treated with rapamycin (**Fig 3C**). Average onset of seizures was also comparable to the nontreated RHEBp.P37L mice (mean ± SEM: 32.6 days ± 2.3; chi-squared (1) = 0.16, $p = 0.69$, log-rank test). Hence, the presence of a heterotopic nodule is not required for RHEBp.P37L mediated seizures, and reducing the formation of these nodules does not always prevent epileptogenesis. When splitting the data of the cortical migration patterns shown in **Fig 3B** for mice with and mice without seizures, a clear correlation was observed between the migration pattern of RHEBp.P37L expressing cells and the presence or absence of seizures: RHEBp.P37L-prenatal treated mice with seizures showed a more severe migration deficit of RHEBp.P37L expressing cells compared to prenatal treated RHEBp.P37L expressing mice that were seizure free (**Fig 3D**). In fact, the percentage of cells that reached L2/3 of the SScx of RHEBp.P37L-prenatal treated mice with seizures (63%) was significantly lower than RHEBp.P37L-prenatal treated mice without seizures (93%) or control mice (98%) (% targeted cells in L2/3: H(2) = 22.08, $p < 0.0001$, Kruskal–Wallis test; empty vector versus RHEBp.P37L-no seizures, $p > 0.99$; empty vector versus RHEBp.P37L-seizures, $p < 0.0001$; RHEBp.P37L-no seizures versus RHEBp.P37L-seizures, $p = 0.002$; Dunn's multiple comparisons test). These results indicate that ectopic cells facilitate the process of epileptogenesis.

Hyperactivation of mTORC1 is sufficient to cause seizures, independent of the presence of cortical malformations [27], even when the mTORC1 activity is increased in a limited set of neurons [20]. Moreover, the cortical malformation by itself, in the absence of continued mTORC1 signaling, does not cause epilepsy, as was shown by brain-wide inhibition of mTORC1 signaling [20]. To confirm that brain-wide suppression of mTORC1 activity could reduce seizures in our mouse model, we treated a group of mice showing seizures ($N = 6$; 5 to 6 weeks old) systemically for 7 days with the allosteric mTORC1 inhibitor rapamycin

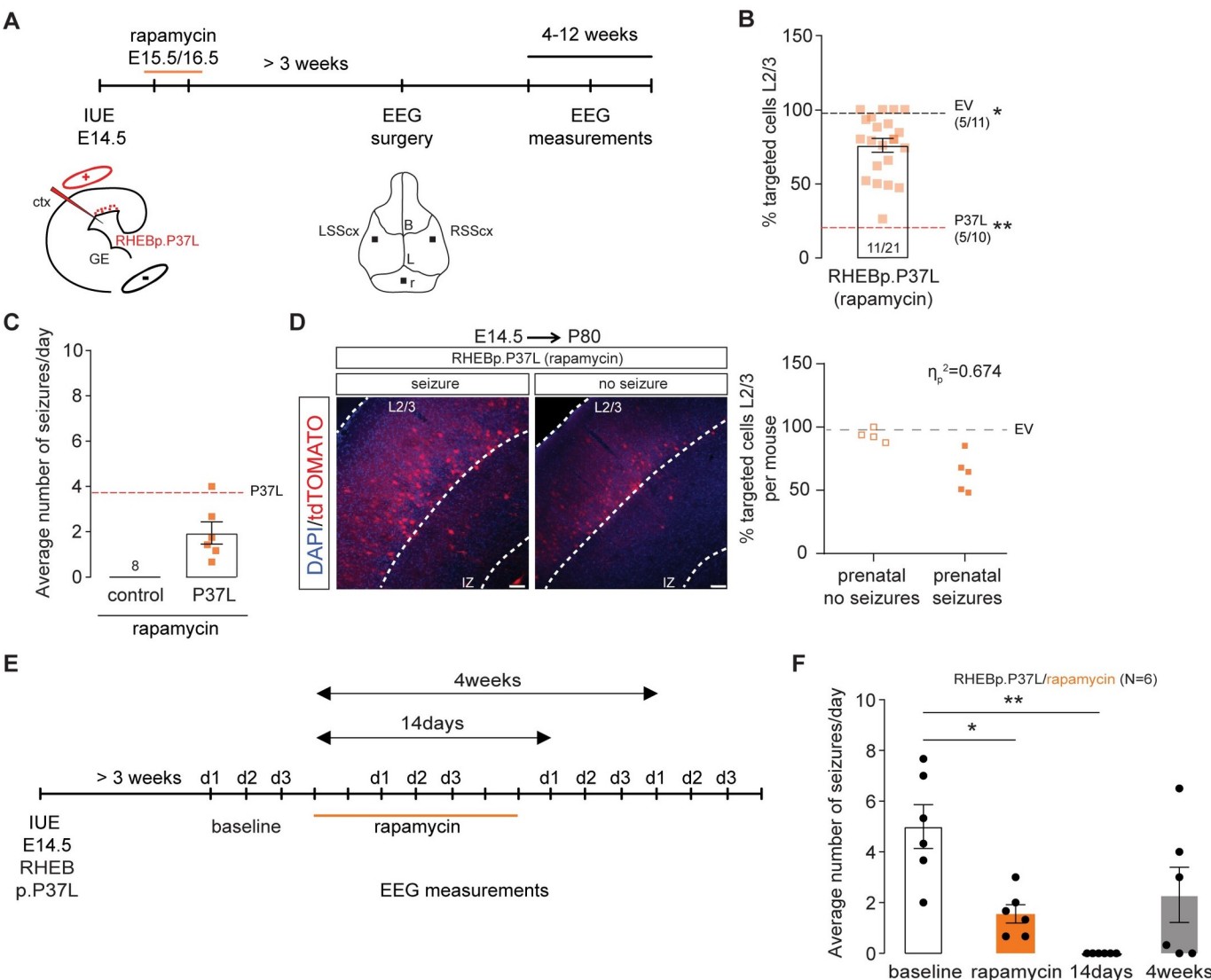

**Fig 3. Rapamycin administration prevents the formation of a heterotopic nodule and stops the occurrence of seizures. (A)** Schematic representation of the timeline of the IUE, SC rapamycin injections, and EEG surgery and measurements. **(B)** Quantification of the percentage of tdTomato+ cells that managed to migrate out to L2/3 in mice prenatally exposed to rapamycin; data are presented as mean ± SEM, single data points represent the values of each picture analyzed and dashed lines indicate the mean value of cells reaching L2/3 in empty vector control mice (black line) and in RHEBp.P37L mice (red line); numbers in the graph indicate number of mice ($N = 11$, $N = 5$) and number of pictures analyzed ($n = 21$, $n = 11$, $n = 10$); (% targeted cells in L2/3: H(2) = 25.97, $p < 0.0001$, Kruskal–Wallis test; EV vs RHEBp.P37L-prenatal rapamycin, $p = 0.05$; RHEBp.P37L vs RHEBp.P37L-prenatal rapamycin, $p = 0.002$, RHEBp.P37L vs EV, $p < 0.0001$, Dunn's multiple comparisons test; $^*p < 0.05$, $^{**}p < 0.01$. **(C)** Average number of seizures per day of each targeted mouse prenatally treated with rapamycin and showing spontaneous seizures measured with EEG between 4 and 12 weeks of age ($N = 6$); control mice are nontargeted mice from the same litters as the RHEBp.P37L mice prenatally exposed to rapamycin ($N = 8$); data are presented as mean ± SEM, the dashed line indicates the average number of seizures of RHEBp.P37L mice not treated with rapamycin (Mann–Whitney $U = 17.5$, $p = 0.09$, two-tailed Mann–Whitney test). **(D)** Representative images of RHEBp.P37L mice prenatally exposed to rapamycin that showed or did not show seizures; the quantification graphs shows the degree of association between the migration phenotype shown in panel B and here presented as % of targeted cells in L2/3(dependent scale variable) for each mouse and the absence or presence of seizures (independent nominal variable) in RHEBp.P37L mice ($N = 4$ and $N = 5$, respectively, with the exclusion of the mice that showed heterotopia); the dashed line represents the mean value of the empty vector control group already shown in B, as comparison; $\eta_p = 0.821$, $\eta_p^2 = 0.674$, Eta measure of association, with values of $\eta_p$ close to one indicating strong association. **(E)** Schematic representation of the timeline of the IUE, systemic rapamycin injections, and EEG measurements performed for 3 consecutive days over different sessions over time. **(F)** Average number of seizures per day of each mouse treated with rapamycin ($N = 6$) measured before treatment (baseline), during and after rapamycin injections; data are presented as mean ± SEM (rapamycin effect over time: F(2.04, 10.19) = 9.1, $p = 0.005$, RM one-way ANOVA; baseline vs rapamycin: $p = 0.03$; baseline vs 14 days: $p = 0.005$; baseline vs 4 weeks: $p = 0.17$; Dunnett's multiple comparisons test; the data underlying this figure can be found in **S3 Data**. $^*p < 0.05$, $^{**}p < 0.01$. Scale bars: 100 μm. B, bregma; ctx, cortex; EEG, electroencephalography; EV, empty vector; GE, ganglion eminence; IUE, in utero electroporation; L, lambda; L2/3, layer 2/3; LSScx, left SScx; r, reference electrode; RM, repeated measures; RSScx, right SScx; SScx, somatosensory cortex.

(10 mg/kg) (**Fig 3E**), which reduced and temporarily abolished the occurrence of seizures within 1 week of the last day of rapamycin administration (**Fig 3F**). However, seizures reoccurred starting 3 weeks after the last injection of rapamycin in 4 out of 6 mice, indicating that sustained inhibition of mTORC1 is required to fully suppress the seizures and that the presence of a heterotopic nodule alone is not sufficient to promote the seizures (**Fig 3F**).

To further confirm the necessity of increased mTOR activity in a subset of cells, irrespective of the presence of a cortical malformation, for the development of epilepsy, we used IUE to focally introduce in the SScx a Lox-Stop-Lox (LSL)-RHEBp.P37L vector or floxed-RHEBp. P37L vector together with a vector expressing the ERT2CreERT2 fusion protein (**Fig 4**). This allowed us to switch RHEBp.P37L expression on or off during different stages of cortical development. IUE of the LSL-RHEBp.P37L construct (**Fig 4A**) in the absence of tamoxifen administration did not result in a migration deficit, or seizures, indicating that the LSL cassette successfully prevented RHEBp.P37L expression (**Fig 4B**). However, once expression of RHEBp.P37L was induced by administration of tamoxifen either at P7 or P21, a subset of the mutant mice (38% of the P7 group and 50% of the P21 group) developed spontaneous seizures (**Fig 4C**), albeit with a delayed onset compared to mice that express RHEBp.P37L throughout development (**S3 Fig**). Expression of the floxed-RHEBp.P37L vector (**Fig 4D**) in the absence of tamoxifen resulted in the development of a heterotopic nodule as well as seizures, as expected (**Fig 4E**). Inducing RHEBp.P37L deletion at P14 (**Fig 4F**) prevented the development of seizures, despite the presence of a heterotopic nodule (**Fig 4E**). Furthermore, inducing deletion of RHEBp.P37L after epileptogenesis completely abolished the seizures within 10 days of gene deletion (*N* = 4, last EEG measurements performed between days 85 and 90) (**Fig 4F**). Taken together, these results confirm that RHEBp.P37L expression in a limited number of cells drives seizure development and that cortical malformations are neither necessary nor sufficient for the development of spontaneous seizures, as shown by the pharmacological data.

## RHEBp.P37L expression induces aberrant axonal development both in vitro and in vivo and increases synaptic connectivity of callosal projecting neurons

The mTOR pathway plays an important role in axonal outgrowth, with functional effects on neuronal network formation [28–30]. Because increased mTOR signaling in a limited number of neurons in the brain is enough to cause seizures, independently from cell misplacement, we hypothesized that this could be due to aberrant neuronal connectivity caused by RHEBp.P37L overexpression. Therefore, we investigated the effect of RHEBp.P37L expression on axonal length and branching both in vitro and in vivo. Overexpression of RHEBp.P37L in primary hippocampal neurons in vitro caused a significant increase in axonal length and axonal branching, compared to the empty vector control (**Fig 5A**). In vivo, axons from callosal projection neurons originating from the superficial layers of the SScx project to the homotopic contralateral hemisphere, where they mostly innervate L2/3 and L5 pyramidal neurons [31,32]. They also send collaterals to L2/3 and, more strongly, L5 pyramidal neurons within the same column ipsilaterally, participating in local circuitry [32,33]. Therefore, it is conceivable that in vivo overexpression of RHEBp.P37L affects callosal projections to the nontargeted contralateral SScx hemisphere. Analysis of the contralateral callosal axonal growth in matched coronal sections with comparable targeting revealed that upon RHEBp.P37L overexpression, axonal terminals in the contralateral hemisphere showed a broader distribution compared to controls, reaching the primary (S1) and secondary (S2) SScx (**Fig 5B**). Furthermore, a significant difference was found in the distribution of the axonal terminals across the different layers in the contralateral hemisphere. In the control condition, most of the terminals in the contralateral

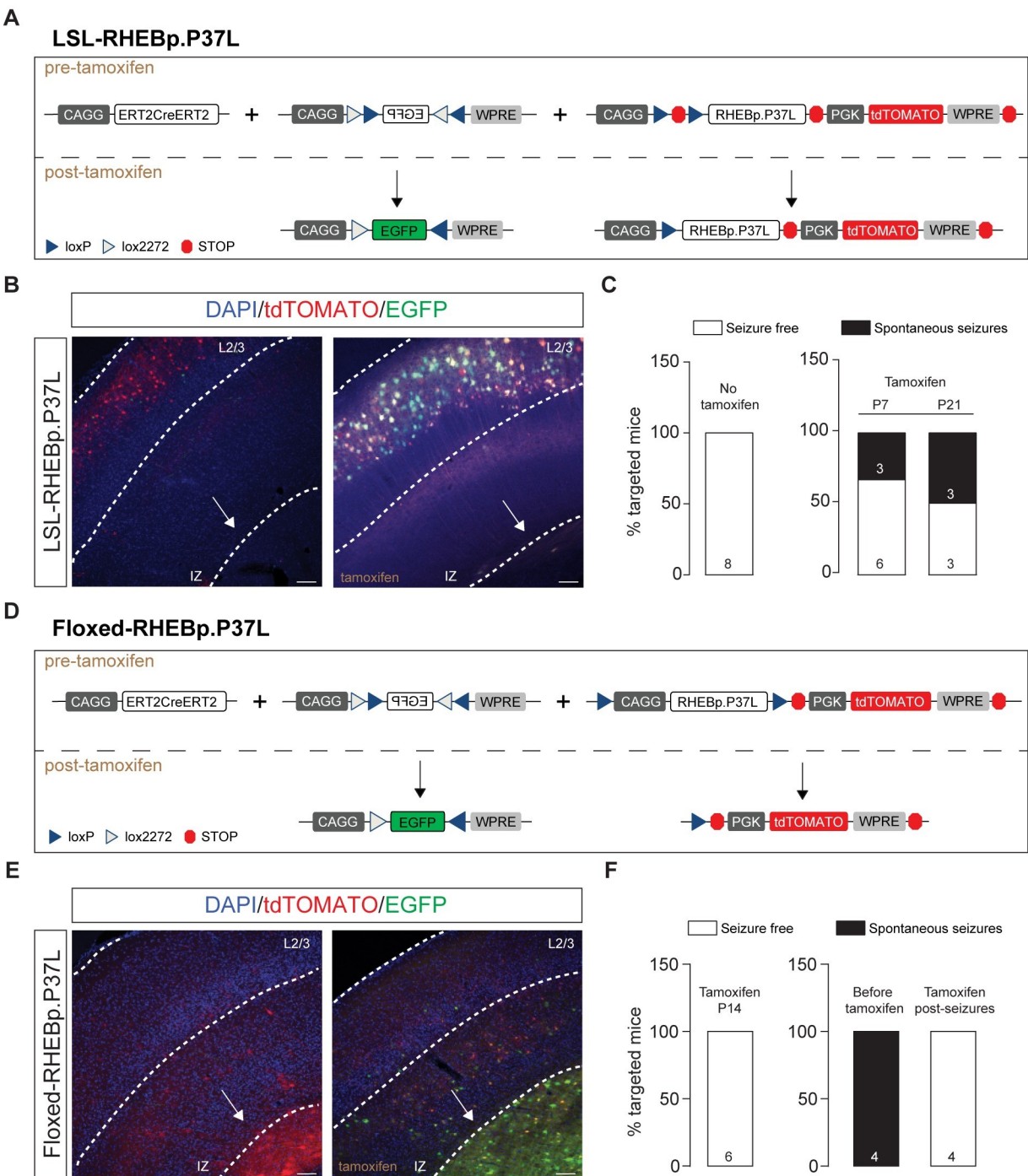

**Fig 4. The heterotopic nodule is neither necessary nor sufficient to induce spontaneous seizures. (A and D)** Schematic representation of the DNA plasmids used in the experiment. The LSL or the floxed construct was expressed in combination with the CAGG-ERT2CreERT2 and a CAGG-DIO-EGFP constructs. The EGFP in the CAGG-DIO-EGFP construct is expressed only upon tamoxifen injection, providing a measure of efficient cre-dependent recombination (see representative images in **B** and **E**). **(B)** Representative images showing efficient cre recombination upon tamoxifen administration in adult mice injected in utero with the LSL construct; note the absence of heterotopic nodule (indicated by the white arrow). **(C)** Bar graphs indicating the percentage of targeted mice showing seizures after injection at either P7 or P21 for 4 consecutive times with tamoxifen and measured with EEG until 12 weeks of age; numbers in the bar plots indicate the number of mice. **(E)** Representative images showing efficient cre recombination upon tamoxifen administration in adult mice injected in utero with the floxed construct; note the presence of heterotopic nodule (indicated by the white arrow). **(C)** Bar graphs indicating the percentage of targeted mice showing seizures after injection at either P14 or upon seizure development for 4 consecutive times with tamoxifen and measured with EEG until 12 weeks of age; numbers in the bar plots indicate the number of mice. Scale bars: 100 μm. The data underlying this figure can be found in **S4 Data**. EEG, electroencephalography; LSL, Lox-Stop-Lox.

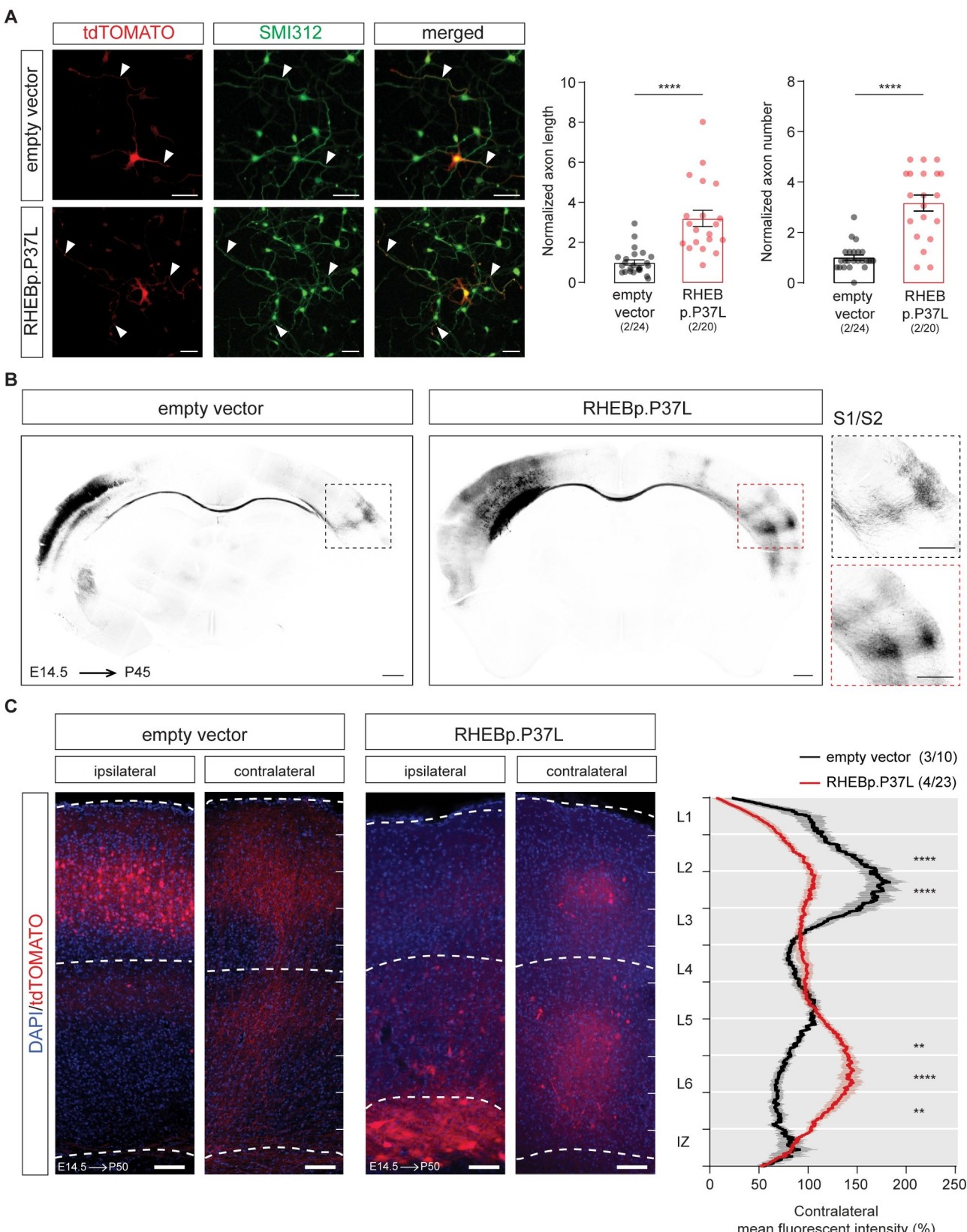

**Fig 5. RHEBp.P37L overexpression induces an increase in axon length and branching both in vitro and in vivo. (A)** Representative images of primary hippocampal cultures transfected at DIV1 with either empty vector control or RHEBp.P37L constructs (tdTomato, in red) stained at DIV4 with a pan axonal marker SMI312 (in green); arrowheads indicate the axons; bar graphs represent mean ± SEM and single data points indicate the number of cells analyzed; numbers indicate number of neuronal cultures ($N = 2$) and total number of cells analyzed ($n = 24$, $n = 20$); axonal length: Mann–Whitney $U = 32$, $p < 0.0001$, Mann–Whitney test; axonal branches: Mann–Whitney $U = 53$, $p < 0.0001$, Mann–Whitney test. **(B)** Overview coronal sections in grey scale stained with anti-RFP antibody of an empty vector and a RHEBp.P37L mouse brain in utero electroporated on the left S1 and magnification of the axon terminals on the contralateral S1; scale bars: 500 μm. **(C)** Representative images of ipsilateral and contralateral S1 area of an empty vector and a RHEBp.P37L mouse coronal section (P50) with quantification of the axonal projections across the different layers in the contralateral cortex measured as normalized fluorescent intensity of the tdTomato signal; numbers in the legend indicate number of targeted mice ($N = 3$, $N = 4$) and number of contralateral pictures ($n = 10$, $n = 23$) analyzed; data are presented as mean (thick line) ± SEM (shading area); interaction group condition/cortical layers: $F(9, 279) = 13.96$, $p < 0.0001$, mixed-effects analysis; control vs RHEBp.P37L L2/3 (bin2–3 from the top): $p < 0.0001$; control vs RHEBp.P37L L5-L6: bin7, $p = 0.0074$, bin8, $p < 0.0001$, bin 9, $p = 0.002$; Bonferroni multiple comparisons test. The data underlying this figure can be found in **S5 Data**. **$p < 0.01$, ****$p < 0.0001$; scale bars: 50 μm **(A)**, 500 μm **(B)**, and 100 μm **(C)**. DIV1, 1 day in vitro; DIV4, day in vitro 4.

SScx were located in L2/3, with a lower abundance in L5 [31]. In the RHEBp.P37L mice, we found that most of the terminals were located in the deeper layers of the SScx, suggesting an improper cortical connectivity (**Fig 5C**). Furthermore, zooming in on the axonal projections on the contralateral cortex of RHEBp.P37L mice revealed the presence of enlarged boutons and terminals positive for Synapsin-1 and VGLUT1, markers for synaptic vesicles and gluta-matergic neurons, respectively (**S4A Fig**). Electron Microscopy (EM) analysis of these boutons confirmed that the boutons were on average twice the size of control boutons and filled with neurotransmitter vesicles, which potentially could result in increased connectivity compared to control (**S4B Fig**).

To investigate if the contralateral axonal projections with synaptic terminals showing altered morphology are functional and indeed show increased connectivity, we made use of optogenetics. We used IUE to introduce channelrhodopsin-2 (pCAGGS-ChR2-Venus) [32] together with either the empty vector control or the RHEBp.P37L construct in targeted neurons and recorded the postsynaptic responses (excitatory postsynaptic potentials (EPSCs)) to widefield optogenetic stimulation by patch-clamping L2/3 and L5 pyramidal neurons in the (nontargeted) contralateral S1 where axonal terminals could be observed (**Fig 6A** and **S5A Fig**). Analyzing the amplitude of EPSCs following optogenetic stimulation in L5 and L2/3 of the contralateral S1, we observed an overall increase in response in the RHEBp.P37L condition compared to the empty vector control condition (see **S3 Table** for statistics) (**Fig 6B**). When analyzing the total charge of the compound postsynaptic response, we observed similar response patterns (**S3 Table** for statistics) (**Fig 6B**). Bath application of tetrodotoxin (TTX) in the RHEBp.P37L group decreased the postsynaptic responses evoked by photostimulating ChR2 expressing fibers to noise level, which is indicative of action potential-driven neuro-transmitter release (**S5B Fig**). The basic properties (resting membrane potential [Vm] and membrane resistance [Rm]) of L2/3 and L5 contralateral cells in empty vector control and RHEBp.P37L conditions were not different (**S3 Table** for statistics). These data suggest increased synaptic connectivity to the contralateral S1 upon overexpression of RHEBp.P37L.

## Loss of axonal projections or blocking vesicle release of RHEBp.P37L expressing neurons is sufficient to stop seizures

Having shown that the RHEBp.P37L expressing neurons show stronger axonal innervation and synaptic connectivity to neurons in the contralateral hemisphere, we investigated whether these altered neuronal projections drive the seizures. To assess this, we made use of the tetanus toxin light chain, known to specifically cleave the SNARE complex protein Synaptobrevin/VAMP2 (Syb2) [34]. VAMP2 is part of the SNARE complex that allows synaptic vesicles fusion and the release of neurotransmitters [35], and recently, it has been shown to mediate the vesicular release of brain-derived neurotrophic factor (BDNF) from axon and dendrites, thereby

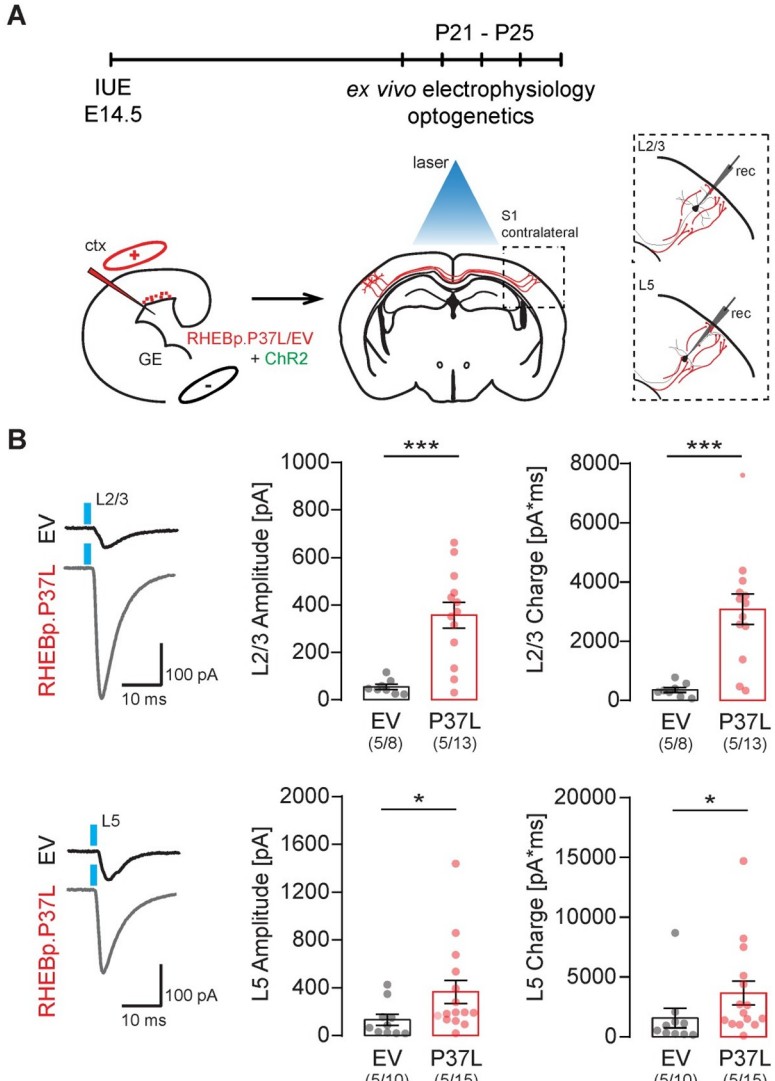

**Fig 6. Overexpressing RHEBp.P37L increases synaptic connectivity on the contralateral hemisphere. (A)** Schematic representation of the timeline and experimental conditions of the IUE and ex vivo whole-cell patch clamp recordings in contralateral L2/3 and L5 upon widefield optogenetic stimulation (indicated by the blue laser). **(B)** Example traces and analysis of the compound postsynaptic responses after photostimulation (blue light), showing the postsynaptic response amplitudes and total charge in contralateral L2/3 and L5 in EV and RHEBp.P37L expressing slices; numbers in the graph indicate number of targeted mice ($N = 5$) and number of cells ($n = 8$, $n = 13$, $n = 10$, $n = 15$) analyzed; data are presented as mean ± SEM and single data points indicate the values of each cell; for statistics see **S3 Table**; the data underlying this figure can be found in **S6 Data**. $^{*}p < 0.05$, $^{***}p < 0.001$. ctx, cortex; EV, empty vector; GE, ganglion eminence; IUE, in utero electroporation; L2/3, layer 2/3.

regulating proper cortical connectivity [36]. Intrinsic neuronal activity during early brain development is crucial for axonal growth and branching, and blocking synaptic transmission using tetanus toxin interferes with proper cortical axonal formation, resulting in the reduction and disappearance of axonal projections [37]. Indeed, when RHEBp.P37L was cotransfected with a tetanus toxin construct (TeTxLC) that is active during embryonic development, we observed a complete block of callosal axonal growth in the contralateral SScx (**Fig 7A**). Furthermore, the mice targeted with the RHEBp.P37L and TeTxLC constructs did not develop

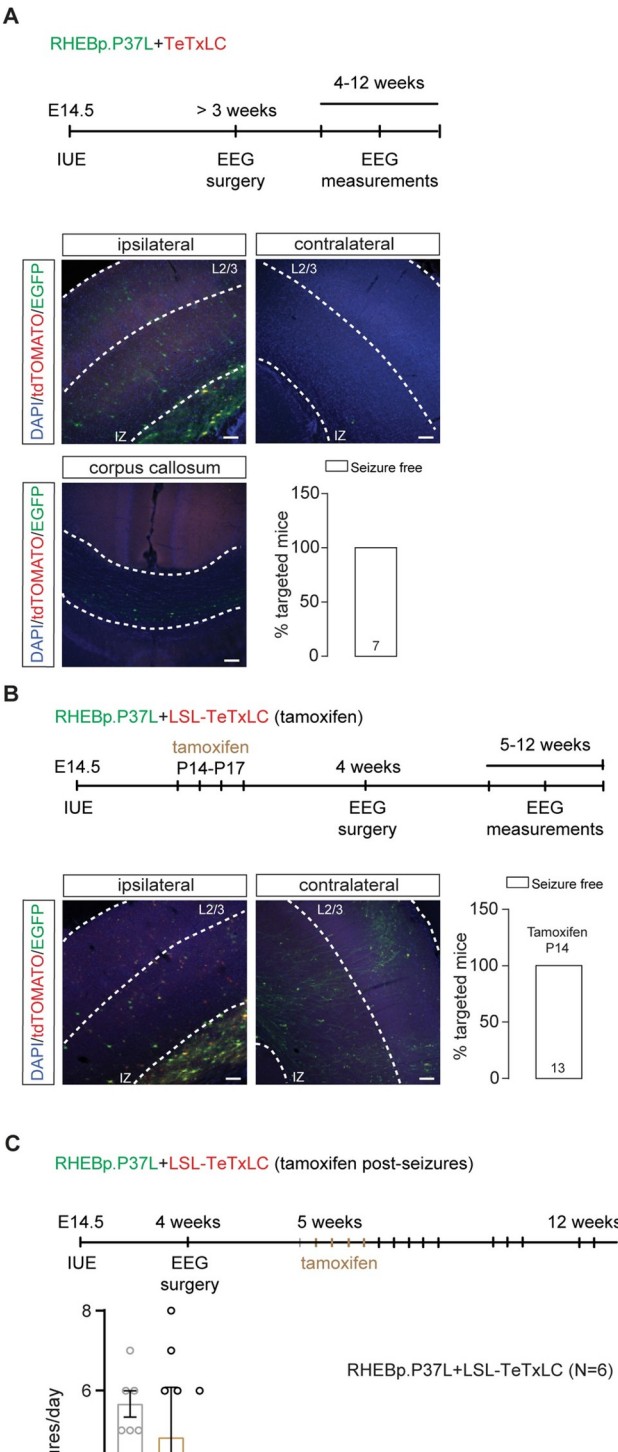

**Fig 7. Loss of axonal projections or blocking vesicle release of RHEBp.P37L expressing neurons is sufficient to stop seizures. (A)** Schematic representation of the timeline of the IUE, EEG surgery, and EEG measurements upon expression of a RHEBp.P37 (construct expressing EGFP, in green) and a TeTxLC (construct expressing tdTomato, in red). Example figures of ipsilateral targeted S1, corpus callosum, and contralateral S1 of an in utero electroporated adult mouse at 12 weeks of age. Note the absence of axonal projections on the contralateral side. The bar graph shows percentage of seizure-free targeted mice measured with EEG until 12 weeks of age. Numbers in the bar graph indicate number of mice. **(B)** Schematic representation of the timeline of the IUE, tamoxifen administration, EEG surgery, and EEG measurements upon expression of a RHEBp.P37L (construct expressing EGFP, in green) and a LSL-TeTxLC (in red). Example figures of ipsilateral targeted area (left) and contralateral cortex of an in utero electroporated adult mouse (12 weeks) injected with tamoxifen starting at P14 for 4 times. The bar graph shows percentage of targeted mice developing seizures upon early tamoxifen injection (P14) and measured with EEG until 12 weeks of age. Numbers in the bar graphs indicate number of mice. **(C)** Schematic representation of the timeline of the IUE, EEG surgery and measurements, and tamoxifen administration upon expression of a RHEBp.P37L (construct expressing EGFP, in green) and a LSL-TeTxLC (in red). Tamoxifen was administered for 4 consecutive days after seizures were first measured for full 24 hours, and mice were continuously monitored with EEG for 10 consecutive days. Mice were measured again at week 8/9 and finally at week 11/12 before sacrifice. Data are presented as mean ± SEM, and each data point represents a single mouse (tamoxifen effect over time: F(2.8, 14.20) = 17.9, $p$ < 0.0001, RM one-way ANOVA). The data underlying this figure can be found in **S7 Data.** Scale bars: 100 μm. EEG, electroencephalography; IUE, in utero electroporation; IZ, intermediate zone; L2/3, layer 2/3; LSL-TeTxLC, LSL-tetanus toxin light chain; TeTxLC, tetanus toxin light chain.

seizures, suggesting that the abnormal axonal connectivity might mediate the expression of seizures in our mouse model (**Fig 7A**).

The complete loss of callosal axonal branching upon embryonic activation of TeTxLC prevented us from testing whether increased synaptic transmission drives seizure development. Therefore, to enable activation of the tetanus toxin upon tamoxifen injection at postdevelopmental stages, we generated an inducible LSL-TeTxLC construct and cotransfected this construct with RHEBp.P37L and the CAGG-ERT2CreERT2 vector (see **Fig 4A**). This allowed us to assess whether, once (abnormal) axonal projections are established, blocking vesicular release either prevents the development of seizures or stops seizures once they have developed. Activation of the tetanus toxin during postnatal development, but before seizure onset (P14), completely prevented the development of seizures while allowing the axons to grow and branch to the contralateral side (**Fig 7B**). Administering tamoxifen in 5-week-old mice, when the cortical connectivity is complete and after the mice showed seizures, revealed that epilepsy is not an irreversible process (**Fig 7C**). Already after 2 days of tamoxifen administration, 3 out of 6 mice stopped showing any seizures, and 2 weeks after the last tamoxifen injection, all mice appeared to be seizure free (**Fig 7C**). These results indicate that inhibiting synaptic transmission by blocking vesicular release from the targeted cells is enough to stop the occurrence of seizures in our mouse model.

## Neurons in the contralateral homotopic cortical area in RHEBp.P37L expressing mice show increased excitability

To obtain more insight into the cellular mechanisms that underlie epilepsy in our model, we used whole-cell patch clamp to measure intrinsic physiological properties of the RHEBp.P37L expressing neurons, of (ipsilateral) neurons directly surrounding the targeted cells, and of the contralateral neurons in homotopic cortical areas (**Fig 8A**). Whole-cell patch clamp recordings were performed by recording from pyramidal neurons in S1 of 3-week-old mice. For the RHEBp.P37L expressing neurons (tdTomato positive), we recorded from neurons that managed to migrate out to L2/3 of S1 to be able to compare their physiological properties with "empty vector" control cells in L2/3 that expressed the tdTomato gene without expressing the RHEBp.P37L protein (**Fig 8A**). RHEBp.P37L expressing neurons showed an increase in the capacitance (*Cm*) compared to empty vector control cells (**Fig 8B** and see **S4 Table** for

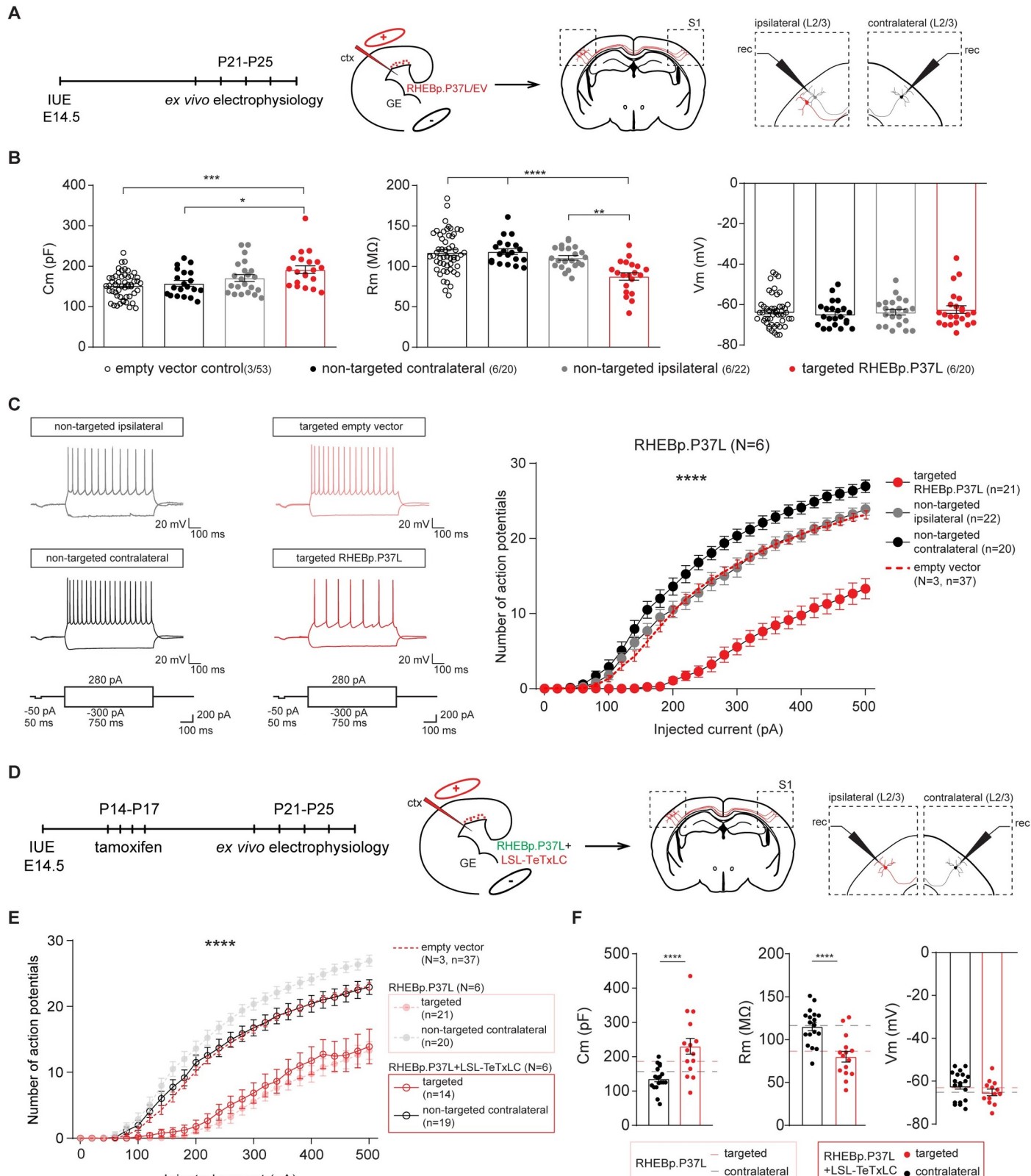

**Fig 8. Neurons in the contralateral homotopic cortical area in RHEBp.P37L mice show increased excitability that can be reversed by blocking vesicles release.**
**(A)** Schematic representation of the timeline and experimental conditions of the IUE and ex vivo whole-cell patch clamp recordings showing the targeted cells

patched in the targeted S1, L2/3, and nontargeted L2/3 cells in the ipsilateral and contralateral sides. **(B)** Analysis of the passive membrane properties (Cm, Rm, and Vm) of pyramidal cells in L2/3 (targeted and nontargeted) of control EV mice and targeted and nontargeted pyramidal cells in L2/3 of RHEBp.P37L mice; numbers in the legend indicate number of targeted mice ($N = 3$, $N = 6$) and number of cells ($n = 37$, $n = 20$, $n = 22$, $n = 21$) analyzed; data are presented as mean ± SEM and single data points indicate the values of each cell; for statistics, see **S4 Table**. **(C)** Example traces and number of action potentials in response to increasing depolarizing currents; number of mice and cells is as indicated in **(B)**; data are presented as mean ± SEM and the red dashed line represents the pooled mean value ± SEM of targeted and nontargeted cells in empty vector control mice ($N = 3$) shown separately in **S5A Fig**, for comparison; for statistics, see **S4 Table**. **(D)** Schematic representation of the timeline and experimental conditions IUE, tamoxifen injections, and ex vivo whole-cell patch clamp recordings in L2/3 of ipsilateral and contralateral S1 cortex. **(E)** Number of action potentials in response to increasing depolarizing currents of cells expressing both RHEBp.P37L and LSL-TeTxLC in L2/3 ipsilateral S1 and nontargeted cells in L2/3 contralateral S1; data are presented as mean ± SEM and dashed lines represent the mean values ± SEM of the pooled control cells from EV mice shown in **S5A Fig** and of the RHEBp.P37L mice from **Fig 8C**, for comparison; $N$ = number of mice, $n$ = number of cells analyzed; for statistics see **S5 Table**. **(F)** Analysis of passive membrane properties (Cm, Rm, and Vm) of pyramidal cells in L2/3 of mice targeted with RHEBp.P37L and LSL-TeTxLC in ipsilateral S1 and nontargeted cells on the contralateral side; data are presented as mean ± SEM and the dashed lines indicate the mean values of Cm, Rm, and Vm of RHEBp.P37L targeted cells in L2/3 and contralateral cells shown in **Fig 8B**, for comparison; for statistics see **S5 Table**. The data underlying this figure can be found in **S8 Data**. *$p < 0.05$, **$p < 0.01$, ***$p < 0.001$, ****$p < 0.0001$. Cm, capacitance; ctx, cortex; EV, empty vector; GE, ganglion eminence; IUE, in utero electroporation; L2/3, layer 2/3; LSL-TeTxLC, LSL-tetanus toxin light chain; Rm, membrane resistance; Vm, resting membrane potential.

statistics), which is consistent with the increase in soma size (median of control empty vector cells L2/3: 1.005, n cells = 22; median RHEBp.P37L cells L2/3: 1.377, n cells = 24; Mann–Whitney $U = 105$, $p = 0.0003$, two-tailed Mann–Whitney test). Additionally, the membrane resistance ($Rm$) was decreased, whereas the resting membrane potential ($Vm$) was unchanged compared to empty vector control cells (**Fig 8B** and see **S4 Table** for statistics). Depolarizing the neurons with increasing current injections showed that the excitability of cells expressing the empty vector were not different from nontargeted neurons in the same mice or compared to nontargeted mice (**S6A Fig**). In contrast, RHEBp.P37L expressing neurons were hypoexcitable compared to control neurons measured in mice expressing the empty vector as well as to nontargeted neurons ipsilateral and contralateral (**Fig 8C** and see **S4 Table** for statistics), without a change in the threshold $Vm$ to fire action potentials (F (3, 94) = 0.59, $p = 0.62$, nonsignificant, one-way ANOVA). This result is again in agreement with the observed increased soma size and concomitant increased cell capacitance and decreased membrane resistance. RHEBp.P37L expressing neurons located in the nodule behaved similarly to the RHEBp.P37L-positive neurons that managed to migrate out to L2/3. Indeed, they exhibited a similar level of hypoexcitability (**S6B Fig**) and besides a small increase in membrane capacitance ($Cm$) displayed similar passive properties to the L2/3 RHEBp.P37L-positive neurons (**S6C Fig**). This suggests once more that the location of these cells is not the primary determinant of the physiological behavior of RHEBp.P37L expressing neurons. Notably, while ipsilateral nontransfected neurons surrounding the RHEBp.P37L expressing neurons in mice did not show changes in excitability compared to empty vector control, nontransfected neurons in L2/3 on the contralateral hemisphere showed a significant increase in excitability (**Fig 8C** and see **S4 Table** for statistics), suggesting that the ectopic cells affect long-range connected neurons.

To experimentally address if the aberrant connectivity could cause the increase in excitability in neurons on the contralateral cortex, we again made use of the inducible LSL-TeTxLC construct and cotransfected this construct with RHEBp.P37L and the CAGG-ERT2CreERT2 vector to enable activation of the tetanus toxin upon tamoxifen injection at postdevelopmental stages (See **Fig 7B**). Whole-cell patch clamp recordings revealed that activating the tetanus toxin early during development (P14) (**Fig 8D**) completely reversed the hyperexcitability phenotype of the contralateral nontargeted cells observed in the RHEBp.P37L mice (**Fig 8E**), while the targeted cells cotransfected with RHEBp.P37L and the tetanus toxin maintained the hypoexcitable phenotype and the basic properties observed in the RHEBp.P37L group (**Fig 8E and 8F** and see **S5 Table** for statistics). Taken together, these data indicate that the abnormal axonal connectivity caused by RHEBp.P37L overexpression is the primary driver of the hyperexcitability phenotype of contralateral L2/3 pyramidal neurons, which, in turn, could be the main driver of epilepsy.

## Discussion

In this study, we investigated the mechanisms behind the spontaneous tonic–clonic seizures in a mouse model for mTOR-related MCD, generated by spatially and temporally restricted over-expression of an mTOR pathway-related ID mutation in *RHEB* [23]. We showed that the RHEBp.P37L mutant is resistant to inhibition by the TSC complex and that restricted overexpression causes mTORC1 hyperactivity and the development of heterotopia with typical cellular features of human MCD such as enlarged dysplastic neurons with altered morphology and mTORC1 activation. Furthermore, the presence of cortical malformations is accompanied by the development of spontaneous tonic–clonic seizures and alterations of cortical brain dynamics that are rescued by administration of rapamycin, an mTOR inhibitor. Using pharmacological and genetic approaches we showed that the mTOR-dependent epilepsy likely arises from altered axonal connectivity, through which distally connected (supposedly healthy) neurons are affected. Indeed, blocking either mTOR activity or vesicle release from the relatively few neurons in which mTOR is hyperactive is enough to stop or prevent seizures.

Similar to previously generated IUE mouse models of MCD, our model developed clear heterotopia, strikingly resembling human focal cortical malformations, associated with mTORC1 hyperactivity and reliable spontaneous seizures [19–21,38,39]. The malformation in our mouse model is characterized by white matter heterotopia and neuronal misplacement across the different cortical layers but maintains the molecular fingerprint belonging to L2/3 neurons. However, it is difficult to categorize it as a specific type of MCD because it expresses characteristics of both FCD type I and type IIa (with no balloon cells observed) [1]. Nonetheless, the targeted cells have features common to several types of mTOR-dependent MCD, including enlarged and dysplastic cells with mTORC1 hyperactivation [15].

Previously, it has been shown that brain-wide activation of the mTOR pathway is sufficient to induce seizures in the absence of any cortical malformations [27]. However, these models do not address the role of mTOR signaling in MCD-related pathophysiology. To address that, an elegant IUE mouse model was generated, which expressed the constitutively active RHEBp.S16H protein. These mice showed a migration deficit resembling FCD and spontaneous epilepsy [20]. Using this mouse model, it was also shown that the presence of a cortical malformation is not necessary to induce seizures [20]. Notably, these mice did not show epilepsy when the SScx was targeted, and hence, the investigators suggested that the SScx might be a nonepileptogenic area. This is in contrast with our mouse model using the human ID-related RHEBp.P37L mutant and with several studies that manipulate other components of the mTOR pathway, where targeting the SScx reliably induces seizures [38–42].

Our mouse model offers a good tool to test novel antiepileptic drugs (AEDs) in vivo. However, considering the variability in the number of seizures exhibited, it will be beneficial to focus on different parameters when assessing the potential therapeutic efficiency of AEDs. For this purpose, the *theta* frequency oscillation, which we found to be affected and normalized upon rapamycin treatment, represents a good biomarker for assessing the potential therapeutic value of treatments in our mouse model [25,26].

Everolimus and rapamycin (Sirolimus) have been shown in randomized controlled trials to be beneficial for treating TSC-associated epilepsy [43,44] but not for treating the cognitive deficits [45,46]. In this study, we investigated the potential of a short prenatal rapamycin treatment in improving both malformation defects and epilepsy but preventing the possible side effects (developmental delays and poor gain weight) [47]. We showed that a 2-day rapamycin treatment during a critical time point of prenatal development can cause a substantial improvement of the cortical malformation defects and prevents the development of seizures in almost 50% of the cases. Future studies will have to assess if a combination of prenatal and

postnatal treatment with rapamycin in mice can be sufficient to significantly reduce the epileptic events, as shown for brain malformations, without causing major side effects [47,48].

Surgery is often an alternative to AEDs for treating MCD-related epilepsy. Human electrophysiological findings show that seizures can often have multiple starting points, besides the brain lesion itself [49,50]. Therefore, from a clinical point of view, it is important to determine whether seizures originate from cells surrounding the cortical malformation. Even though EEG and LFP do not have the spatial resolution to assess the primary epileptogenic zone in our model, we showed that persistent mTORC1 hyperactivation in the targeted cells is the primary cause of epilepsy. In fact, genetically removing the RHEBp.P37L mutant, either before or after seizure development, was sufficient to prevent or stop the epilepsy.

Surprisingly, when exploring the causes of epileptogenesis, we observed that the neurons expressing the RHEBp.P37L both in layer 2/3 and in the heterotopic nodule are hypoexcitable, which is consistent with the increase in soma size but does not provide an obvious physiological explanation for the seizures observed in our mouse model. This is in sharp contrast with human studies that show that cytomegalic neurons from resected brain tissue of MCD patients are hyperexcitable and might therefore play a central role in the generation of epileptic discharges [51–53]. However, we observed a clear increase of intrinsic excitability and in postsynaptic responses upon optogenetic stimulation of RHEBp.P37L cells in contralateral homotopic S1 cells. This suggests that RHEBp.P37L expressing cells induce cellular changes in anatomically connected neurons, which might underlie, or at least exacerbate, the epilepsy phenotype. This would further support the hypothesis that the interaction between abnormal and normal cells in the brain results in an epileptic focus and might explain the insurgence of epilepsy in more mild cases of MCD [51,54–56]. Notably, the alterations we observed in our model extend well beyond the cells surrounding the cortical malformation, as we found physiological changes were present contralateral to the targeted side. Considering the abnormal axonal connectivity seen in our mouse model, this raises the possibility that other anatomically connected cortical and subcortical areas not analyzed in this study might also be affected, thereby providing an explanation for how a small percentage of targeted hypoexcitable cells, independent of their location, can lead to generalized epilepsy. Therefore, we propose a model in which subtle microscopic alterations and aberrant connectivity, either through an increase in synaptic connections or an increase in the strength of synaptic contacts caused by mTOR hyperactivity, are sufficient to drive epileptogenesis.

By increasing axonal connectivity, RHEBp.P37L expressing neurons could potentially alter synaptically connected neurons through neurotransmitter release. But they can also affect neighboring (including synaptically nonconnected) cells through the release of extracellular vesicles such as exosomes [57]. The vesicles might mediate pathogenicity as was previously shown in vitro [58]. Our results show that although most electrophysiological parameters of the ipsilateral nontargeted cells are unaltered compared to control, there is a small trend for an increased membrane capacitance (Cm) in these cells (see **Fig 8B**). Whether this is caused by a non-cell autonomous effect, or a secondary effect due to presence of the seizures, remains to be investigated. With the use of tetanus toxin, we showed that the effects on the contralateral side are directly driven by the abnormal enhanced axonal connectivity, since blocking vesicle release specifically from the RHEBp.P37L expressing neurons completely rescued the epilepsy and normalized the intrinsic firing properties of the nontargeted contralateral neurons. Tetanus toxin is primarily used to block synaptic transmission due to its effect on neurotransmitter release, acting on the SNARE complex protein VAMP2 [34]. Given the observed increased axonal connectivity and the finding that distally connected cells were physiologically affected, this strongly suggests that neurotransmitter-mediated communication is primarily causing the epilepsy phenotype. This notion is further supported by the optogenetics experiments that

showed increase postsynaptic responses upon stimulating the RHEBp.P37L expressing neurons. While it has been proposed that specific tetanus-insensitive VAMP proteins (such as VAMP7) are involved in the release of exosomes into the extracellular space [59], we cannot exclude the additional contribution of other types of vesicles to the observed phenotype. Recently, it was shown that tetanus toxin–sensitive SNAREs also drive the release of BDNF [36]. Some studies suggest that BDNF might contribute to epileptogenesis [60], suggesting that abnormal BDNF signaling could further increase the epileptic phenotype seen in our mouse model. Understanding the contribution of these different signaling pathways is important for the development of targeted therapeutic strategies to treat MCD-associated epilepsy.

In summary, here, we made use of a hyperactive RHEB mutant that was previously identified in patients with ID, megalencephaly, and epilepsy, as a model for human mTOR-related MCD-associated epilepsy. We show that in this model, a few neurons with increased mTOR activity can be the driving force behind MCD-related epilepsy through aberrant connectivity, resulting in increased excitability of distant nontargeted neurons, which can be reversed by blocking vesicular release.

## Materials and methods

### Mice

Unless subjected to a surgical procedure, all experimental mice were kept group-housed in IVC cages (Sealsafe 1145T, Tecniplast) with bedding material (Lignocel BK 8/15 from Rettenmayer) on a 12/12-hour light/dark cycle at 21°C (±1°C), humidity at 40% to 70%, and with food pellets (801727CRM(P) from Special Dietary Service) and water available ad libitum. For the neuronal cultures, FvB/NHsD females were crossed with FvB/NHsD males (both ordered at 8 to 10 weeks old from Envigo). For the IUE, females FvB/NHsD (Envigo) were crossed with males C57Bl6/J (ordered at 8 to 10 weeks old from Charles River). Both females and males from the IUE litters were included in the experiments, and no prescreening for successful electroporation was performed before recruitment in the studies. Young (starting from P7) and adult mice were used, and the specific age for each experiment is indicated either in the results section or in the figures' legends. Activation of the ERT2CreERT2 fusion protein [61] was achieved by intraperitoneal administration of tamoxifen for 4 consecutive days (0.1 mg/g of bodyweight) dissolved in sunflower oil (20 mg/ml) at the ages specified in the results section and in the figures. For inhibition of the mTOR pathway, rapamycin (Sigma-Aldrich, St Louis, USA) was dissolved in dimethylsulfoxide (10 mg/ml) and injected intraperitoneally in adult mice (>4 weeks) for postnatal experiments (10 mg/kg) or subcutaneously in pregnant females (E15.5/E16.5) for prenatal experiments (1 mg/kg).

All animal experiments were conducted in accordance with the European Commission Council Directive 2010/63/EU (CCD project license AVD1010020172684), and all described experiments and protocols were subjected to ethical review (and approved) by an independent review board (IRB) of the Erasmus MC (workprotocol numbers: 17-2684-01, 17-2684-03, and 17-2684-04).

### HEK293T cell cultures and transfection

HEK293T cells were grown in Dulbecco's modified Eagle medium (DMEM; Lonza, Verviers, Belgium) supplemented with 10% fetal bovine serum, 50 U/ml penicillin, and 50 µg/ml streptomycin in a 5% $CO_2$ humidified incubator at 37°. Before transfection, $1 \times 10^5$ HEK293T cells were seeded per well of 24-well culture dishes and transfected 24 hours later with expression constructs encoding the *RHEB* variants (0.2 µg), the *S6K* reporter (0.2 µg), *TSC1* (0.2 µg), and *TSC2* (0.2 µg) using Lipofectamine 2000 (Invitrogen, Carlsbad, California, USA). To ensure

that a total of 0.8 μg plasmid DNA was added per well, empty pcDNA3 vector was included where necessary. The day after transfection, the growth medium was replaced with DMEM without glucose and incubated for a further 4 hours prior to harvesting and western blot analysis.

## Western blotting

After transfection, HEK293T cells were transferred on ice, washed with PBS (4°C), and lysed in 70 μl 50 mM Tris-HCl (pH 7.6), 100 mM NaCl, 50 mM NaF, 1% Triton X100 in the presence of protease and phosphatase inhibitors (Complete, Roche Molecular Biochemicals, Woerden, the Netherlands). Cell lysates were subjected to immunoblotting using the following primary antibodies: anti-RHEB mouse monoclonal [62], anti-TSC1 and TSC2 rabbit polyclonal [63], T389-phosphorylated S6K (1A5, #9206, Cell Signaling Technology), and rabbit anti-myc (#2272, Cell Signaling Technology), all 1:1,000. Primary antibody binding was assessed by incubation with goat anti-rabbit (680 nm) and anti-mouse (800 nm) conjugates (1:15,000, Li-Cor Biosciences, Lincoln, USA) followed by detection on an Odyssey near-infrared scanner (Li-Cor Biosciences).

## Neuronal primary hippocampal cultures and transfection

Primary hippocampal neuronal cultures were prepared from FvB/NHsD wild-type mice according to the procedure described in [64]. Neurons were transfected at 1 day in vitro (DIV1) with the following DNA constructs: control empty vector (1.8 μg per coverslip) and RHEB p.P37L (2.5 μg per coverslip). Plasmids were transfected using Lipofectamine 2000 according to the manufacturer's instructions (Invitrogen, Waltham, USA).

## Plasmids

cDNA encoding the *RHEB* (NM_005614.3) c.110C>T (p.P37L) mutation was synthesized by GeneCust. The c.46-47CA>TG (p.S16H) variant was generated by site-directed mutagenesis (Invitrogen) using the following primers: Fw 5′–gcgatcctgggctaccggCAtgtggggaaatcctcatt– 3′ and Rev 5′–aatgaggatttccccacaTGccggtagcccaggatcgc– 3′. All *RHEB* gene variants were cloned in our dual promoter expression vector using AscI and PacI restriction sites [23], and the empty vector used as control refers to the dual promoter expression vector without a gene inserted and expressing either tdTOMATO or EGFP (specified in the figures or in the figures' legends). Expression constructs for *TSC1*, *TSC2*, and a myc-tagged *S6K* reporter were as described previously [65]. The following DNA plasmids were obtained from Addgene: pGEMTEZ-TeTxLC (Addgene plasmid #32640; http://n2t.net/addgene:32640; RRID: Addgene_32640) [66]; RV-CAG-DIO-EGFP (Addgene plasmid #87662; http://n2t.net/addgene:87662; RRID:Addgene_87662) [67]; pCAG-ERT2CreERT2 (Addgene plasmid #13777; http://n2t.net/addgene:13777; RRID:Addgene_13777) [61]; pCAGGS-ChR2-Venus (Addgene plasmid #15753; http://n2t.net/addgene:15753; RRID:Addgene_15753) [32]. The TeTxLC was isolated by PCR using the following primers: Fw 5′–taagcaggcgcgccaccatgccgat-caccatcaacaa– 3′ and Rev 5′–gccatggcggccgcgggaattcgat– 3′ and inserted in our dual promoter expression vector using AscI and NotI restriction sites. To generate the loxP-STOP-loxP (LSL) constructs (loxP-STOP-loxP-*RHEB* p.P37L and loxP-STOP-loxP-TeTxLC), the LSL sequence was obtained from the Ai6 CAG-Floxed ZsGreen in Rosa 26 targeting vector (Addgene plasmid #22798; http://n2t.net/addgene:22798; RRID:Addgene_22798) using multiple cloning sites and inserted just after the CAGG promoter and before the beginning of the gene in our dual promoter expression vector containing either *RHEB*p.P37L or TeTxLC. The floxed *RHEB* p.P37L construct was generated by introducing 2 loxP site sequences before the CAGG

promoter and at the end of the *RHEB*p.P37L gene, with the same orientation to ensure proper deletion. To achieve this, the following couples of oligonucleotides were used for annealing: Fw 5′- cgcgtATAACTTCGTATAGCATACATTATACGAAGTTATg—3′, Rev: 5′- ctagcATA ACTTCGTATAATGTATGCTATACGAAGTTATa—3′; Fw: 5′- taaATAACTTCGTATAG CATACATTATACGAAGTTATg—3′, Rev: 5′- tcgacATAACTTCGTATAATGTATGCTA TACGAAGTTATttaat—3′.

## In utero electroporation

IUE was performed as described previously [68]. Pregnant FvB/NHsD mice at E14.5 of gestation were used to target the progenitor cells giving rise to pyramidal cells of the layer 2/3. Each *RHEB* cDNA construct (including the LSL and floxed conditions) was diluted to a final concentration of 0.5 μg/μl in fast green (0.05%), while other plasmids were diluted to a concentration of 1.5 to 2 μg/μl. The DNA solution was injected into the lateral ventricle of the embryos while still in utero, using a glass pipette controlled by a Picospritzer III device. When multiple constructs were injected, a mixture of plasmids was prepared to achieve a final concentration of 1.5 to 2 μg/μl, keeping the *RHEB* concentration constant throughout all the experiments. To ensure proper electroporation of the injected constructs (1 to 2 μl) into the progenitor cells, 5 electrical square pulses of 45 V with a duration of 50 ms per pulse and 150 ms interpulse interval were delivered using tweezer-type electrodes connected to a pulse generator (ECM 830, BTX Harvard Apparatus). The positive pole was placed to target the developing SScx. Animals of both sexes were used to monitor seizure development, for ex vivo electrophysiology experiments, or for histological processing with no exclusion criteria determined by a postnatal screening of the targeting area.

## Immunostainings

For immunocytochemistry analysis, neuronal cultures were fixed 3 days posttransfection with 4% paraformaldehyde (PFA)/4% sucrose, washed in PBS, and incubated overnight at 4°C with primary antibodies in GDB buffer (0.2% BSA, 0.8 M NaCl, 0.5% Triton X-100, 30 mM phosphate buffer (PB) (pH7.4)). Mouse pan anti-SMI312 (1:200, BioLegend, #837904) was used to stain for the axon and, after several washings in PBS, donkey anti-mouse-Alexa488 conjugated was used as secondary antibody diluted in GDB buffer for 1 hour at room temperature (1:200, Jackson ImmunoResearch, West Grove, USA). Slides were mounted using mowiol-DABCO mounting medium.

For the staining of brain tissue sections, mice were deeply anesthetized with an overdose of Nembutal and transcardially perfused with 4% PFA in PB. Brains were extracted and postfixed for 1 hour in 4% PFA. They were then embedded in gelatin and cryoprotected in 30% sucrose in 0.1 M PB overnight, frozen on dry ice, and sectioned using a freezing microtome (40 μm thick). Immunofluorescence was performed on free-floating sections that were first washed multiple times in PBS and blocked in 10% normal horse serum (NHS) and 0.5% Triton X-100 in PBS for 1 hour at room temperature. Primary antibodies diluted in PBS containing 2% NHS and 0.5% Triton X-100 were added at room temperature overnight. The day after, sections were washed 3 times with PBS, and secondary antibodies were added diluted in PBS containing 2% NHS and 0.5% Triton X-100. After washing in PBS and 0.05 M PB, sections were counterstained with 4′,6-diamidino-2-phenylindole solution (DAPI, 1:10,000, Invitrogen) before being washed in 0.05 M PB and mounted on slides using chromium (3) potassium sulfate dodecahydrate (Sigma-Aldrich) and left to dry. Finally, sections were mounted on glass with mowiol (Sigma-Aldrich).

Biocytin labelling was achieved by fixating the patched slices overnight in 4% PFA in PB at 4°. Slices were then washed multiple times in PBS and incubated with Alexa488-Streptavidin

(1:200; #016-540-084, Jackson ImmunoResearch) or AlexaCy5-Streptavidin (1:200; #016-170-084, Jackson ImmunoResearch) overnight at 4˚. The next day, after washing in PBS and 0.05 M PB, sections were counterstained with DAPI (1:10,000, Invitrogen) and mounted on glass with mowiol (Sigma-Aldrich).

When performing Nissl stainings, few selected free-floating sections corresponding to the SScx were mounted on glass using chromium (3) potassium sulfate dodecahydrate (Sigma-Aldrich) and left to dry overnight. Slides were stained in 0.1% Cresyl Violet for 4 to 10 minutes, then rinsed briefly in tap water to remove excess stain, dehydrated in increasing percentages of alcohol, cleared with xylene, and covered using Permount (Fisher Scientific, Pittsburgh, USA).

The primary antibodies used in this study to stain for the specific targets indicated for each experiment in the figures' legends were as follows: rabbit anti-pS6 (Ser 240/244), 1:1,000; Cell Signaling, catalog #5634; rabbit anti-RFP, 1:2,000; Rockland, catalog 600-401-379; rabbit anti-RHEB, 1:1,000, Proteintech Group, catalog 15924-1-AP; rabbit anti-CUX1, 1:1,000; Proteintech Group, catalog 11733-1-AP; rat anti-CTIP2, 1:200; Abcam, catalog ab18465; rabbit anti-NeuN, 1:2,000; Millipore catalog ABN78 (RRID: AB_10807945); mouse anti-SATB2, 1:1,000; Santa Cruz, catalog sc-81376; rabbit anti-synapsin 1, 1:1,000; Merck Millipore, catalog #AB1543P; guinea pig anti-VGLUT1, 1:1,000; Merck Millipore, catalog #AB5905; rabbit anti-GABA, 1:500; Sigma-Aldrich, catalog #A2052 and rabbit anti-PV, 1:1,000; Swant, catalog #PV 27; secondary antibodies used were as follows: donkey anti-rabbit 488, catalog #711-545-152; donkey anti-rabbit 647, catalog #711-605-152; donkey anti-rabbit Cy3, catalog #711-165-152; donkey anti-mouse 488, catalog #715-545-150; donkey anti-mouse 647, catalog #715-605-150; donkey anti-rat Cy5, catalog #712-175-150; donkey anti-guinea pig 647, catalog #706-605-148; all from Jackson ImmunoResearch, 1:200.

## Immunoelectron microscopy

Mice in utero electroporated with either empty vector or RHEBp.P37L were anesthetized at P21 with an overdose of nembutal (IP) and transcardially perfused with 10 ml saline and subsequently 50 ml 4% PFA and 0.5% glutaraldehyde in cacodylate buffer. The brain was removed and post-fixed overnight in 4% PFA. Coronal sections (80 µm) were cut on a vibratome (Technical Products International, St. Louis, USA), and sections corresponding to the SScx were further processed for DAB staining. dtTomato-positive terminals were visualized by incubating the sections with the avidin-biotinperoxidase complex (ABC) method for 24 to 48 hours (Vector Laboratories, USA) and subsequently developed with DAB (0.05%, Life Technologies Carlsbad, USA) as the chromogen. The vibratome sections were rinsed and post-fixed in 1% osmium tetroxide, stained with 1% uranyl acetate, and dehydrated and embedded in araldite (Durcupan ACM; Fluka, Buchs, Switzerland). Ultrathin (50 to 70 nm) sections were cut on an ultramicrotome (Leica, Wetzlar, Germany), mounted on formvar-coated copper grids, and contrasted with 2% uranyl acetate and 1% lead citrate (Fluka). The grids were subsequently rinsed twice with TBST and incubated for 1 hour at room temperature in goat anti-rabbit IgG labeled with 10 nm gold particles (Aurion Wagening, The Netherlands) diluted 1:25 in TBST. SScx sections containing tdTomato-positive terminals were photographed using an electron microscope (Philips, Eindhoven, The Netherlands), and the size of the terminals was analyzed using FIJI software.

## LFP and EEG recordings

Starting from 3 weeks of age, surgeries were performed according to the procedures described in [69,70]. After at least 3 days of recovery from the EEG surgical procedure, mice were connected to a wireless EEG recorder (NewBehavior, Zurich, Switzerland) for 24 hours per day for at least 2 consecutive days (1 session of recordings). EEG recordings were manually

assessed by 2 different researchers blind for the genotypes to check for occurrence of seizures, defined as a pattern of repetitive spike discharges followed by a progressive evolution in spike amplitude with a distinct post-ictal depression phase, based on the criteria described in [71]. If no seizures were detected during the first week post-surgery, mice were recorded for another session of 48 to 56 hours for a maximum of 4 sessions over 4 weeks post-surgery. During the days in which no EEG recordings were performed, mice were monitored daily to assess for the presence of behavioural seizures and discomfort.

For the LFP recordings, 2 days after the surgical procedure, mice were head fixed to a brass bar suspended over a cylindrical treadmill to allow anaesthesia-free recording sessions and placed in a light-isolated Faraday cage as described in [70]. Mice were allowed to habituate to the setup before proceeding to the recording. LFP measurements were acquired every day in sessions of 20 to 30 for 5 or 8 consecutive days, using the Open Ephys platform with a sampling rate of 3 kS/s and a band pass filter between 0.1 and 200 Hz. For the power spectrum analysis, the average power density spectrum of all the days of recording was obtained using MATLAB software (MathWorks; RRID:SCR_001622). The mean relative power was calculated over 4 frequency bands relative to the total power: delta (2 to 4 Hz), theta (4 to 8 Hz), beta (13 to 30 Hz), and gamma (30 to 50 Hz).

At the end of each experiment, mice were sacrificed for immunohistological analysis to assess electrodes positioning, amount of targeting, and efficiency of cre-dependent recombination when tamoxifen was administered.

## Ex vivo slice electrophysiology for excitability

P21-P25 mice of both sexes in utero electroporated with the plasmids specified in the figures and in the legends for each experiment were anaesthetized with isoflurane before decapitation. The brain was quickly removed and submerged in ice-cold cutting solution containing (in mM) 110 choline chloride, 2.5 KCl, 1.2 $NaH_2PO_4$, 26 $NaHCO_3$, 25 D-glucose, 0.5 $CaCl_2$, and 10 $MgSO_4$. Acute 300 μm coronal slices were made of the SScx using a vibratome (HM650V, Microm, Thermo Scientific, Walldorf, Germany) while being saturated with 95% $O_2$/5% $CO_2$. The slices were immediately transferred to a submerged slice holding chamber and incubated at ±34˚C for 5 minutes before being transferred to a second slice holding chamber also kept at ±34˚C. The second holding chamber contained the same artificial cerebrospinal fluid (ACSF) as was used during all recordings and contained (in mM) 125 NaCl, 3 KCl, 1.25 $NaH_2PO_4$, 26 $NaHCO_3$, 10 glucose, 2 $CaCl_2$, and 1 $MgSO_4$. During the slicing procedure and experimental recordings, slices were saturated with 95% $O_2$/5% $CO_2$. Slices recovered for an hour at room temperature before starting the experiment. After the experiment, slices were fixed in 4% PFA overnight and then transferred to PBS until further processing. Whole-cell patch clamp recordings were obtained from the soma of visually identified L2/L3 pyramidal neurons from the S1 cortex with an upright microscope using IR-DIC optics (BX51WI, Olympus, Tokyo, Japan). Targeted cells in the ipsilateral side were identified by the presence of either tdTomato or GFP, depending on the experiment, elicited by an Olympus U-RFL-T burner. All recordings were done under physiological temperatures of 30 ± 1˚C. Patch clamp pipettes were pulled from standard wall filament borosilicate glass to obtain electrodes with a tip resistance between 2 and 4 MΩ. All recordings were performed using a Multiclamp 700B (Molecular Devices, Sunnyvale, California, USA) and digitized by a Digidata 1440A (Molecular Devices, Sunnyvale, California, USA). For the current clamp recordings, pipets were filled with a K-gluconate internal solution containing (in mM) 125 K-gluconate, 10 NaCl, 10 HEPES, 0.2 EGTA, 4.5 MgATP, 0.3 NaGTP, and 10 Na-phosphocreatine. For analysis of cell morphology, biocytin (5%) was added to the intracellular solution. The final solution was adjusted to a pH of 7.2 to

7.4 using KOH and had an osmolarity of 280 ± 3. After getting a seal of at least 1 GΩ, whole-cell configuration was obtained by applying brief negative pressure together with a short electric pulse. Prior to breaking in, cell capacitance was compensated. Series resistance was monitored but not corrected. Recordings with an unstable series resistance and higher than 20 MΩ were rejected. Membrane potentials were not corrected for liquid junction potential. Resting membrane potential was measured immediately after break in.

Each sweep started with a small and short hyperpolarizing step (−50 pA, 50 ms) to monitor access resistance. Action potentials were triggered by square step current injections into the patched neurons while holding them at −70 mV. Steps were 750 ms long and started at −300 pA with increments of 20 pA. The number of action potentials and action potential properties were analyzed using Clampfit 10.7.0.3 (Molecular Devices, USA). For each cell, the first action potential at rheobase was analyzed. The threshold was calculated by plotting the first derivative of the trace. The threshold was defined when the first derivative was lower than 10 mV/ms. Series resistance was calculated offline for each cell by plotting the difference in voltages between baseline and the hyperpolarizing steps. A linear line was plotted to visualize passive current only. The tau was calculated by fitting a standard exponential on the end of the hyperpolarizing steps. From tau and series resistance, capacitance was calculated.

## Ex vivo slice electrophysiology for optogenetics

P21-P25 mice of both sexes in utero electroporated either with the RHEBp.P37L and pCAGGS-ChR2-Venus plasmids or the empty vector and pCAGGS-ChR2-Venus plasmids [32] were anaesthetized with isoflurane before decapitation. The brain was quickly removed and submerged in ice-cold cutting solution containing (in mM) 93 NMDG, 93 HCl, 2.5 KCl, 1.2 NaHPO$_4$, 30 NaHCO$_3$, 25 glucose, 20 HEPES, 5 Na-ascorbate, 3 Na-pyruvate, 2 Thiourea, 10 MgSO4, 0.5 CaCl2, and 5 N-acetyl-L-Cysteine (osmolarity 310 ± 5; bubbled with 95% O$_2$/5% CO$_2$) [72]. Next, 250 μm thick coronal slices were cut using a Leica vibratome (VT1000S). For the recovery, brain slices were incubated for 5 minutes in slicing medium at 34 ± 1˚C and subsequently for approximately 40 minutes in ACSF (containing in mM: 124 NaCl, 2.5 KCl, 1.25 Na$_2$HPO$_4$, 2 MgSO$_4$, 2 CaCl$_2$, 26 NaHCO$_3$, and 20 D–glucose, osmolarity 310 ± 5 mOsm; bubbled with 95% O$_2$/5% CO$_2$) at 34 ± 1˚C. After recovery, brain slices were stored at room temperature. For all recordings, slices were bathed in 34 ± 1˚C ACSF (bubbled with 95% O$_2$/5% CO$_2$). Whole-cell patch clamp recordings were recorded with an EPC-10 amplifier (HEKA Electronics, Lambrecht, Germany) and sampled at 20 kHz. Resting membrane potential and input resistance were recorded after whole-cell configuration was reached. Recordings were excluded if the series resistance or input resistance (RS) varied by >25% over the course of the experiment. Voltage and current clamp recordings were performed using borosilicate glass pipettes with a resistance of 3 to 5 MΩ that was filled with K-gluconate-based internal solution (in mM: 124 K-gluconate, 9 KCl, 10 KOH, 4 NaCl, 10 HEPES, 28.5 sucrose, 4 Na$_2$ATP, and 0.4 Na$_3$GTP (pH 7.25 to 7.35; osmolarity 290 ± 5 mOsm)). Recording pipettes were supplemented with 1 mg/ml biocytin to check the location of the patched cells with histological staining. Current clamp recordings were corrected offline for the calculated liquid junction potential of −10.2 mV.

Full-field optogenetic stimulation (470 nm peak excitation) was generated by the use of a TTL-pulse controlled pE2 light emitting diode (CoolLED, Andover, UK). Light intensities at 470 nm were recorded using a photometer (Newport 1830-C equipped with an 818-ST probe, Irvine, California) at the level of the slice. To trigger neurotransmitter release from targeted axons, we delivered a 1-ms light pulse with an intensity of 99.8 mW/mm$^2$ at a frequency of 0.1 Hz. To ensure that we recorded action potential-driven neurotransmitter release, most

experiments were concluded by bath application of 10 μM TTX, which blocked all postsynaptic responses in the recorded pyramidal neurons.

## Imaging and analysis

Images of Nissl stained sections were acquired in brightfield with a Nanozoomer scanner (Hamamatsu, Bridgewater, New Jersey) at a 40X resolution using the NDP.view2 software. All immunofluorescent images were acquired using a LSM700 confocal microscope (Zeiss, Oberkochem, Germany). For the analysis of the axons in vitro, at least 10 distinct confocal images from 2 different neuronal batches were taken from each coverslip for each experiment (20X objective, 0.5 zoom, 1024 × 1024 pixels; neurons were identified by the red immunostaining signal). The simple neurite tracer plugin from the FIJI ImageJ software was used for the analysis of the axonal length and branches. Overview images of the coronal sections were acquired by tile scan with a 10X objective. Zoom in images of the targeted area (ipsilateral) and contralateral S1 were taken using a 10X objective. For the migration analysis, confocal images (10X objective, 0.5 zoom, 1024 × 1024 pixels) were taken from 2 to 3 nonconsecutive sections from at least 2/3 electroporated animals per condition. Images were rotated to correctly position the cortical layers, and the number of cells in different layers was counted using the "analyze particles" plugin of FIJI. The results were exported to a spreadsheet for further analysis. Cortical areas from the pia to the ventricle were divided into 10 bins of equal size, and the percentage of tdTomato-positive cells per bin was calculated. The counting of the number of targeted cells per mouse was performed by selecting 3 nonconsecutive (250 to 300 μm apart) targeted sections, with 1 section being the most targeted and the other 2 immediately frontal and caudal from this. The total number of tdTomato+ cells per section was quantified, and the average per mouse calculated to obtain a representative value corresponding to the amount of targeting per mouse. The soma size analysis was performed on z-stacks images acquired using a 20X objective, 1 zoom, 1024 × 1024 pixels of the targeted cells in both empty vector control and *RHEB*p.P37L coronal sections. A ROI around each targeted cell in maximum intensity projection pictures was defined using the FIJI software, and the area of the soma was measured using the "Measure" option in ImageJ. The analysis of the size of EM boutons was performed using the FIJI software. For the analysis of pS6 intensity levels, confocal images (10X objective, 0.5 zoom, 1024 × 1024 pixels) of the ipsilateral and contralateral S1 cortex were acquired with the same master gain from both control and RHEB groups previously stained together against pS6 (240/244). The overall intensity level of the staining for each picture was measured using the "RGB measure" plugin of FIJI, and the values of each ipsilateral side were normalized against the corresponding contralateral side and plotted as averaged values. The analysis of the fluorescent intensity of the axonal branches over the contralateral cortical layers was obtained from 3 to 4 matched coronal sections from at least 3 different animals per group with comparable amount of targeting. The axonal arborization was measured selecting the S1/S2 border, drawing a straight segmented line with adjusted width and length and resized in 1,000 bins, and using the "plot profile" option of the analyze section of FIJI to measure the fluorescent intensity of the tdTomato signal over the different layers. The values obtained for each section were exported to a spreadsheet where they were normalized against the mean background fluorescent intensity calculated on a nontargeted, cortical area of fixed size and plotted as averaged values over 10 bins of equal size. For the analysis of the morphology of biocytin-filled pyramidal cells and ectopic cells in the nodule labelled with streptavidin-488 or streptavidin-Cy5, z-stacks images were taken using a 20X objective, 0.5 zoom, 1024 × 1024 pixels to include the dendritic tree. Maximum intensity projection pictures were analyzed using the SynD software for the MATLAB platform to automatically detect the dendritic morphology and perform Sholl analysis [73].

## Statistics

Normality of the distribution for the different experiments was determined using either the Wilk–Shapiro test or the Kolmogorov–Smirnov test. Statistical analysis was performed using a one-way ANOVA (or corresponding nonparametric Kruskal–Wallis test), two-way repeated measures ANOVA or mixed-effects analysis, Student $t$ test (or corresponding nonparametric Mann–Whitney test), and correlation/association analysis. The specific test used for each experiment and relative significance are specified in the figures' legends, in the supporting information tables, or in the results section. For all statistical analyses, α was set at 0.05. Values are represented as average ± SEM or as median, minimum, and maximum values (specified in the figures' legends). No samples or mice were excluded from the final analysis. Group sizes, biological replicates, number of cells, samples, or brain sections are indicated in the figures and their corresponding legends. All statistical tests were performed either using GraphPad Prism 8.0 (RRID: SCR_002798) or SPSS Statistics v25.0 (RRID:SCR_002865).

## Supporting information

**S1 Fig. Cells overexpressing the RHEBp.P37L construct show enlarged soma size, maintain the molecular identity of pyramidal cells L2/3, and show mTOR hyperactivity. (A)** Nissl staining of coronal brain sections from 5-week-old mice shows the presence of a clear heterotopia (indicated by the empty arrow) in the WM of RHEBp.P37L targeted SScx compared to the empty vector control situation. Boxes a and b represent magnifications of layer 2 (a) and the heterotopia (b), highlighting the targeted dysplastic and enlarged cells (indicated by the arrows); scale bars: 100 μm. **(B)** Representative overview images of coronal sections (SScx) of empty vector control and RHEBp.P37L targeted mice (5 weeks old) probed with common cortical layers markers CUX1 (L2/3 marker), CTIP2 (L5 marker), SATB2 (cortical projection neuron marker), or NeuN (mature neuron marker). **(C)** Representative images of coronal sections (SScx) of RHEBp.P37L targeted mice (5 weeks old) probed with GABA and PV markers for interneurons; magnification pictures of the heterotopia in (c) and (d) show that GABA- and PV-positive cells (indicated by the white arrows) are not positive for tdTomato **(D)** overview of the targeted SScx of empty vector control and RHEBp.P37L targeted mice stained for pS6-240, a readout of mTOR activity. Scale bars: 50 μm. Hc, hippocampus; IZ, intermediate zone; L2/3, layer 2/3; mTOR, mammalian target of rapamycin; SScx, somatosensory cortex; WM, white matter.
(TIF)

**S2 Fig. RHEBp.P37L mice show spontaneous seizures and alterations in the LFP gamma frequency band. (A)** Onset of seizure activity for the RHEBp.P37L group (mean ± SEM: 33.33 days ± 3.26; N indicates number of mice). **(B)** Average number of seizures per day of mice showing seizure activity measured with EEG until 9–12 weeks of age; each data point represents the average per mouse measured at least over 2 separate sessions of recordings (3 days each). **(C)** Simple scatter correlation graph with best fit regression line (Y = −9.5*X + 278.1), showing no correlation between the average number of targeted cells (measured over 3 anatomically matched nonconsecutive targeted slices per mouse) and the average number of seizures per animal shown in figure (B); r(10) = −0.41, $p$ = 0.19, two-tailed Pearson's correlation. **(D)** Extended normalized PSD shown in Fig 3E to include the *beta* and *gamma* frequencies (till 50 Hz); data are presented as mean (thick lines) ± SEM (shadows); N in the legend indicates number of mice per group. **(E)** Quantification of the *beta* (13–30 Hz) and *gamma* (30–50 Hz) frequency bands over the total power; box plots represent minimum and maximum value with median; N in the legend indicates number of mice per group. See **S2 Table**

for statistics; the data underlying this figure can be found in **S2 Data**. $^{**}p < 0.01$. EEG, electro-encephalography; LFP, local field potential; ns, non-significant; PSD, power spectrum density.
(TIF)

**S3 Fig. Delayed seizure development in LSL-RHEBp.P37L mice injected with tamoxifen.** Onset of seizure activity for the LSL-RHEBp.P37L groups after treatment with tamoxifen (4 injections) starting at either P7 (purple line, mean ± SEM: 50 days ± 0) or at P21 (yellow line, mean ± SEM: 74.6 days ± 4.37) compared to the RHEBp.P37L group (red line) (chi-squared (2) = 25.33, $p < 0.0001$; log-rank test; N indicates number of mice). The data underlying this figure can be found in **S4 Data.**
(TIF)

**S4 Fig. Synaptic boutons in RHEBp.P37L mice have an altered morphology and are bigger in size. (A)** Representative zoomed in pictures of the contralateral S1 (L2/3 and L5) of both control EV mice and RHEBp.P37L mice (P50); note the presence of enlarged terminals and boutons in RHEBp.P37L expressing cells that are positive for Synapsin-1 (a marker for synaptic vesicles, in green) and VGLUT1 (a marker for glutamatergic neurons, in green). Scale bars: 10 µm (overview), 5 µm (boutons). **(B)** Representative EM pictures of contralateral S1 boutons of control EV mice and RHEBp.P37L mice (P21) and quantification of the size, showing increase in size in the RHEBp.P37L mice (Mann–Whitney $U = 465$, $p < 0.0001$, $^{****}$, two-tailed Mann–Whitney test); numbers in the graph indicate number of animals/number of boutons analyzed. Scale bars: 500 nm. The data underlying this figure can be found in **S5 Data**. EV, empty vector; L2/3, layer 2/3.
(TIF)

**S5 Fig. Action potentials driven neurotransmitter release in RHEBp.P37L/ChR2 expressing fibers. (A)** Representative images showing expression of ChR2 (in green) and either EV (left) or RHEBp.P37L (right) constructs in red (tdTomato+ cell) on the ipsilateral targeted S1; examples of contralateral patched cells in either L2/3 or L5 filled with byocitin and stained with streptavidin-Cy5 are shown for each condition and indicated with arrowheads (note that for the contralateral pictures, ChR2-Venus is not shown and green represents byocitin-Cy5); scale bars: 100 µm. **(B)** Wash-in of TTX in RHEBp.P37L slices proves the action potential dependence of photostimulation evoked responses in L2/3 and L5; t(5) = 4.8, $p = 0.005$, two-tailed paired $t$ test; $^{**}p < 0.01$. The data underlying this figure can be found in **S6 Data.** ChR2, channelrhodopsin-2; EV, empty vector; TTX, tetrodotoxin.
(TIF)

**S6 Fig. Excitability phenotype of control cells from empty vector control and nontargeted mice and physiological characterization of cells targeted with the RHEBp.P37L in the heterotopic nodule. (A)** Number of action potentials in response to increasing depolarizing currents shows that there is no difference in excitability in empty vector targeted mice or nontargeted mice; data are presented as mean ± SEM; interaction injected current/group condition: F (75, 1225) = 0.7275, nonsignificant; mixed-effects analysis; $N$ = number of mice and $n$ = number of cells analyzed. **(B)** Number of action potentials in response to increasing depolarizing currents shows that there is no difference in excitability in cells targeted with RHEBp.P37L based on their location (L2/3 and nodule); data are presented as mean ± SEM; interaction injected current/group condition: F (25, 824) = 0.95, nonsignificant; mixed-effects analysis; $N$ = number of mice and $n$ = number of cells analyzed. **(C)** Analysis of the passive membrane properties (Cm, Rm, and Vm) of pyramidal cells in L2/3 and cells in targeted cells in the nodule of RHEBp.P37L mice; note the increase in Cm of the targeted cells in the nodule, suggesting a bigger soma size compared to L2/3 cells; Cm: t(32) = 3.6; $p = 0.001$, two-tailed

unpaired $t$ test; Vm: t(35) = 1.02; $p$ = 0.31, two-tailed unpaired $t$ test; Rm: t(32) = 1.11; $p$ = 0.28, two-tailed unpaired $t$ test; numbers in the legend indicate number of targeted mice (N) and number of cells (n) analyzed; data are presented as mean ± SEM, and single data points indicate the values of each cell. The data underlying this figure can be found in **S8 Data.** Cm, capacitance; L2/3, layer 2/3; Rm, membrane resistance; Vm, resting membrane potential. (TIF)

**S1 Data. Numerical data underlying Fig 1.** Excel spreadsheet containining in separate sheets the numerical data underlying Fig 1A, 1C, 1D, 1E, and 1G, respectively. (XLSX)

**S2 Data. Numerical data underlying Fig 1 and S2 Fig.** Excel spreadsheet containining in separate sheets the numerical data underlying Fig 2E and 2F and S2D and S2E Fig, S2A, S2B and S2C Fig, respectively. (XLSX)

**S3 Data. Numerical data underlying Fig 3.** Excel spreadsheet containining in separate sheets the numerical data underlying Fig 3B, 3C, 3D, and 3F, respectively. (XLSX)

**S4 Data. Numerical data underlying Fig 4 and S3 Fig.** Excel spreadsheet containining in separate sheets the numerical data underlying Fig 4C–4F and S3 Fig, respectively. (XLSX)

**S5 Data. Numerical data underlying Fig 5 and S4 Fig.** Excel spreadsheet containining in separate sheets the numerical data underlying Fig 5A and 5C (in separate seets for empty vector control and RHEBp.P37L and combined together) and S4B Fig, respectively. (XLSX)

**S6 Data. Numerical data underlying Fig 6 and S5 Fig.** Excel spreadsheet containining in separate sheets the numerical data underlying Fig 6B and S5B Fig, respectively. (XLSX)

**S7 Data. Numerical data underlying Fig 7.** Excel spreadsheet containining the numerical data underlying Fig 7A–7C. (XLSX)

**S8 Data. Numerical data underlying Fig 8 and S6 Fig.** Excel spreadsheet containining in separate sheets the numerical data underlying Fig 8B, 8C–8E and 8F and S6A–S6C Fig, respectively. (XLSX)

**S1 Raw Images. Uncropped western blots data related to Fig 1A.** (TIF)

**S1 Table. Statistical analysis related to Fig 1.** The table summarizes the statistical tests and values obtained upon analysis of the data presented in Fig 1A. (DOCX)

**S2 Table. Statistical analysis related to Fig 2 and S2 Fig.** The table summarizes the statistical tests and values obtained upon analysis of the data presented in Fig 2E and 2F and S2D and S2E Fig. (DOCX)

**S3 Table. Statistical analysis related to Fig 6.** The table summarizes the statistical tests and values obtained upon analysis of the data presented in Fig 6.
(DOCX)

**S4 Table. Statistical analysis related to Fig 8B and 8C.** The table summarizes the statistical tests and values obtained upon analysis of the data presented in Fig 8B and 8C.
(DOCX)

**S5 Table. Statistical analysis related to Fig 8E and 8F.** The table summarizes the statistical tests and values obtained upon analysis of the data presented in Fig 8E and 8F.
(DOCX)

# Acknowledgments

The authors would like to thank Elize Haasdijk, Charlotte de Konink, Hang Le Ha, Anne Polman, and Chi Yeung for their technical contribution and Minetta Elgersma-Hooisma for managing the mouse colony. We also would like to thank Enzo Nio for advice regarding the code used to analyze the data and Steven Kushner and Dick Jaarsma for fruitful discussions.

# Author Contributions

**Conceptualization:** Martina Proietti Onori, Ype Elgersma, Geeske M. van Woerden.

**Formal analysis:** Martina Proietti Onori, Linda M. C. Koene, Carmen B. Schäfer, Mark Nellist.

**Funding acquisition:** Ype Elgersma, Geeske M. van Woerden.

**Investigation:** Martina Proietti Onori, Linda M. C. Koene, Carmen B. Schäfer, Mark Nellist.

**Methodology:** Martina Proietti Onori, Ype Elgersma, Geeske M. van Woerden.

**Software:** Marcel de Brito van Velze.

**Supervision:** Zhenyu Gao, Geeske M. van Woerden.

**Visualization:** Martina Proietti Onori.

**Writing – original draft:** Martina Proietti Onori, Geeske M. van Woerden.

**Writing – review & editing:** Martina Proietti Onori, Linda M. C. Koene, Carmen B. Schäfer, Mark Nellist, Marcel de Brito van Velze, Zhenyu Gao, Ype Elgersma, Geeske M. van Woerden.

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
