## [Editor Report · Decision Letter 0]

24 Aug 2020

Dear Dr van Woerden, 

Thank you for submitting your manuscript entitled "RHEB/mTOR-hyperactivity causing cortical malformations drives seizures through increased axonal connectivity" for consideration as a Research Article by PLOS Biology.

Your manuscript has now been evaluated by the PLOS Biology editorial staff as well as by an academic editor with relevant expertise and I am writing to let you know that we would like to send your submission out for external peer review.

Please re-submit your manuscript within two working days, i.e. by Aug 26 2020 11:59PM.

Kind regards,

Lucas Smith, Ph.D.,

Associate Editor

PLOS Biology

---

## [Decision Letter · Decision Letter 1]

19 Oct 2020

Dear Dr van Woerden,

Thank you very much for submitting your manuscript "RHEB/mTOR-hyperactivity causing cortical malformations drives seizures through increased axonal connectivity" for consideration as a Research Article at PLOS Biology. Your manuscript has been evaluated by the PLOS Biology editors, an Academic Editor with relevant expertise, and by several independent reviewers.

The reviews of your manuscript are appended below. As you will see from their detailed responses, the reviewers have noted that the study is well conducted and presents a useful model. However they have raised a number of specific concerns which will need to be addressed in order for your manuscript to be considered for publication at PLOS Biology. For example, Reviewer 1 would like you to confirm whether RHEB-p.P37L-cortical cells are excitatory neurons and requests additional characterization of GABAergic interneurons in pathological regions. Reviewer 2 notes a number of locations that require additional clarification, discussion of relevant literature, and additional analyses to be performed to strengthen the study. Reviewer 3 would like the mutant RHEB (RhebP37L) to be compared to a WT RHEB control, instead of a vector alone. Reviewer 3 also asks whether survival is affected in mice expressing RhebP37L, and whether these animals exhibit behavior seizures.

In light of these reviews, we will not be able to accept the current version of the manuscript, but we would welcome re-submission of a much-revised version that takes into account the reviewers' comments. We cannot make any decision about publication until we have seen the revised manuscript and your response to the reviewers' comments. Your revised manuscript is also likely to be sent for further evaluation by the reviewers.

We expect to receive your revised manuscript within 3 months. 

**IMPORTANT - SUBMITTING YOUR REVISION**

*Re-submission Checklist*

*Published Peer Review*

*PLOS Data Policy*

*Blot and Gel Data Policy*

Sincerely,

Lucas Smith, Ph.D.,

Associate Editor,

lsmith@plos.org,

PLOS Biology

REVIEWS:

Reviewer's Responses to Questions

PLOS authors have the option to publish the peer review history of their article (what does this mean?). If published, this will include your full peer review and any attached files.

Reviewer #1: No

Reviewer #2: No

Reviewer #3: Yes: Mauro Costa-Mattioli

Reviewer #1: The manuscript biologically presented epileptogenesis of RHEB-p.P37L-derived cortical cells. Hyperactive mTOR-system is well-known as a pathogenesis of FCD, HME and other brain malformation with intractable epilepsy. The authors showed the evidence of abnormal cell to cell connection and autonomous excitability. This model will be useful for understanding the pathogenesis and therapeutic hint of various brain malformation-related epilepsy. 

I have to require the following correction and consideration.

1. As you know, malformed brain usually demonstrates abnormal interneuron distribution. And, the balance of excitatory neuron and inhibitory interneuron is sometimes discussed. You showed cell autonomous epileptogenesis of RHEB-p.P37L-cortical cells. RHEB-p.P37L-cortical cells might be speculated to be excitatory neurons. Therefore, I would like to request to confirm whether RHEB-p.P37L-cortical cells were excitatory neurons or GABAergic interneurons. Then, I additionally request the distribution of GABAergic interneurons in the pathological region. 

2. I cannot understand Figure 3D. The number of tdTOMATO+ cells in the no-seizure cortex showed less than that in the seizure cortex in the figure panels. However, in the graph the average %cells in the no-seizure cortex showed less than that in the seizure cortex . I feel the results were discrepancy.

Reviewer #2: Authors present a study focused on mTOR activator RHEB-related MCD. They investigated an IUE mouse model (previously published) expressing the p.P37L variant and show it resembles the human pathology with migration defects, mTOR activation and epilepsy. 

Authors address key questions in the field on the cause of epileptogenesis and the role of cortical malformations in seizures generation. Among novel finding, they show that RHEB P37L expressing mutant are hypoexcitable, but lead to increase excitability in contralateral neurons due to abnormal axonal connectivity and neurotransmitter release. 

Some data are not novel (neuronal migration defect and spontaneous tonic-clonic seizures in the RHEBp.P37L IUE mouse model for instance). I suggest to better highlight the novel aspects of the study such as abnormal axonal connectivity leading to hyperexcitability of distal neurons. 

Suggestions for improvement:

- In several parts, a better clarification of which brain area is investigated should be provided. For instance, linking the targeted cells to EEG and LFP recordings (please precise ipsi/contralateral side).

- In figure S2C, the authors show no correlation between the number of seizures and the number of targeted cells. How did the authors count this number of cells: is it per section or total number ? It would be interesting to correlate the pS6 intensity of the targeted cells with the number of seizures, as in Nguyen et al., 2019 study.

- In the LSL-Rheb and floxed-Rheb experiments (Figure 4), I suggest to separate the 2 experiments (or rearrange the Figure). Moreover, the authors mention a delayed onset of seizures between their groups and the group with a developmental deletion of Rheb, without showing the data: a survival graph such as in Figure S2A would be appreciated.

- In Figure 5C, the authors show improper cortical connectivity. As the targeted cells in ipsilateral region are misplaced: is there a correlation between the misplacement of the targeted cells (from IZ to layer 2-3) and their contralateral connexions (from deeper to upper layers )?

- A quantification of the bottons in the contralateral connexions may help assess whether synaptic vesicles have a higher density in Rheb-mutated cells, and potentially explain an increase of neurotransmitter release (Figure S3).

- in the optogenetic experiments, it is not clear how the ChR2 were stimulated: in the ipsi-or the contra- lateral part ? Was the ChR2 expression similar between control and Rheb-mutant ? As the Rheb-mutant present potentially increased axonal length and number, it may be useful to normalize the amplitude recorded to this ChR2 expression.

- In the experiment of inducible Tetanus toxin, it would be clearer to separate both experiments: tamoxifen at P14 and tamoxifen after seizures (Figure 7). Moreover, it is not clear when the tamoxifen post-seizure stopped (Figure 7C): was it only 2 days of injections ? In Figure 7E, it would be better to put an indicative age of the animal in day instead of the day after onset. 

- in the last experiment, the authors investigate the electrophysiological properties of targeted, non-targeted ipsi- and non-targeted contra-lateral neurons. However, they only patched neurons which migrated well to L2/3 (based on the fact that control patched neurons are in L2/3), but they also showed that the misplaced neurons (not migrating properly) kept their identity of L2/3 neurons (expressing Cux1). Therefore, it would have been interesting to assess electrophysiological properties of misplaced Rheb neurons to investigate their differences with the ones that migrated well. This should be discussed. 

- In this same experiment, they show that capacitance is increased in targeted Rheb cells compared to control empty vector. But this parameter seems to be similar (not significantly different) between targeted and non-targeted ipsilateral cells. Could the authors discuss this aspect ? Could it be a non-cell autonomous effect ? Indeed, looking at pS6 levels in Figure 1G and S1C, several non-targeted cells seem to have an increased pS6 levels, and a larger soma size.

- in the discussion, the authors mention previous work from the Bordey's lab suggesting that SScx might be an epileptogenic area. It should be stated that other IUE models of mTOR pathway genes do have seizures from the SScx (Baek et al. 2015, D'Gama et al 2017; Lim et al., 2015; Hu et al., 2018, Ribierre et al., 2018, …). 

- Finally, it could be interesting to discuss how their result can be translated to human patients, according to the work of the lab of C. Cepeda.

Reviewer #3: In a recent study (Reijnders et al., 2017), the authors showed that over-expression of RhebP37L mutation leads to defects in neuronal migrations and seizures. In this study, Onori and colleagues further characterize the effect of this mutation in cortical malformations, seizures and axonal connectivity. The authors first show that the RhebP37L mutant is resistant to inhibition by the TSC complex and leads to increased mTORC1 activity, enlarged dysplastic neurons, cortical malformation and seizures. Using a clever approach that combines pharmacology and genetics, the authors that cortical malformation is neither necessary nor sufficient to induce epilepsy. Interestingly, the authors found that the RhebP37L mutation caused enhanced axonal connectivity and increased hyperexcitability. Moreover, blocking vesicle release was enough to prevent seizures in this model.

Overall, I find this to be a well-conducted and well-executed study that will be interested to many investigators in the field. The paper contains a lot of data and the authors should be lauded for their effort. In addition, the study provides strong evidence that cortical malformation and seizures can be dissociated and a potential mechanism (increased axonal connectivity) underlying the pathology associated with mTORopathies. 

 I have a few questions/suggestions:

1. Fig. 1 and throughout the paper. The comparison should be performed between mutant RHEB (RhebP37L) and WT RHEB, instead of vector alone. 

2. Fig. 2. Do the animals exhibit behavior seizures? Is survival affected in mice expressing RhebP37L in the brain?

3. Does the RhebP37L mutant alter mTORC2? If this is the case, does rapamycin treatment prevent the potential increase in mTORC2 activity?

4. Fig. 4. The changes in migration depicted in this figure are very minor and statistics are required to support the author's argument.

5. Fig. 5. Again, WT mTOR should be used as control. Does expression of RhebP37L mutant leads to increased axon number (more than one axon) in vivo?

---

## [Decision Letter · Decision Letter 2]

20 Apr 2021

Dear Dr van Woerden,

Thank you for submitting your revised Research Article entitled "RHEB/mTOR-hyperactivity causing cortical malformations drives seizures through increased axonal connectivity" for publication in PLOS Biology. I have now obtained advice from two of the three original reviewers and have discussed their comments with the Academic Editor. 

As you will see, the reviewers are satisfied with the revision and note that the study provides important new insights into epileptogenesis. Based on the reviews, we will probably accept this manuscript for publication, provided you address the following data and other policy-related requests.

1) ETHICS REQUEST: In the methods section of your manuscript, please include the full name of the IACUC/ethics committee that reviewed and approved the animal care and use protocol/permit/project license. Please also include an approval number (if different from the CCD number that you already provided). 

2) DATA REQUEST: Please provide, as a supplementary file, the data that was used to generate each graph in your manuscript. When generating this file, please make sure to include a legend describing its contents. We will also need you to add a reference to this file in each figure legend (including supplmental). For example, you might say "the data underlying this figure can be found in S1_data. **For more specific information regarding this request, please see the notes below my signature. 

3) DATA REQUEST: Please provide original, uncropped and minimally adjusted images supporting all blot and gel results reported in an article's figures or Supporting Information files. **For more specific information regarding this request and a link to our guidelines, please see the notes below my signature. 

4) We think that the title of your manuscript could be edited slightly to improve clarity and to note the relevance of your work to epilepsy. If you agree, we might suggest changing the title to something like "RHEB/mTOR-hyperactivity causes cortical malformations and epileptic seizures through increased axonal connectivity".

We expect to receive your revised manuscript within two weeks. 

*Published Peer Review History*

*Early Version*

Sincerely,

Lucas Smith, Ph.D.,

Associate Editor,

lsmith@plos.org,

PLOS Biology

ETHICS STATEMENT:

-- Please include the full name of the IACUC/ethics committee that reviewed and approved the animal care and use protocol/permit/project license. Please also include an approval number (if different from the number you already supplied).

DATA POLICY:

Fig 1A,C,E-F; Fig 2E-F; Fig 3B-D,F; Fig 4C,F; Fig 5A,C; Fig 6B; Fig 7A-C; Fig 8B-C,E-F; Fig S2A-E; Fig S3; Fig S4B; Fig S5B; Fig S6A-C

*Please also ensure that figure legends in your manuscript include information on where the underlying data can be found, and ensure your supplemental data file/s has a legend.

*Please ensure that your Data Statement in the submission system accurately describes where your data can be found.

We require the original, uncropped and minimally adjusted images supporting all blot and gel results reported in an article's figures or Supporting Information files. We will require these files before a manuscript can be accepted so please prepare and upload them now. **Please carefully read our guidelines for how to prepare and upload this data: https://journals.plos.org/plosbiology/s/figures#loc-blot-and-gel-reporting-requirements 

Reviewer remarks:

Reviewer #1: The revised manuscript is well corrected and reconstructed. The responses are acceptable.

Reviewer #2 (Stéphanie Baulac): The authors have very well addressed all questions raised by reviewer 2 and 3, and provided a large amount of new experiments.

I would like to congratulate the authors for this remarkable study, bringing important new insights to mTOR-related epileptogenesis.

---

## [Editor Report · Decision Letter 3]

10 May 2021

Dear Dr van Woerden,

On behalf of my colleagues and the Academic Editor, Eunjoon Kim, I am pleased to say that we can in principle offer to publish your Research Article "RHEB/mTOR-hyperactivity causes cortical malformations and epileptic seizures through increased axonal connectivity" in PLOS Biology, provided you address any remaining formatting and reporting issues. These will be detailed in an email that will follow this letter and that you will usually receive within 2-3 business days, during which time no action is required from you. Please note that we will not be able to formally accept your manuscript and schedule it for publication until you have made the required changes.

When addressing these last formatting requests, we also ask that you address the following two lingering editorial requests: 

1) In the methods section, please include the approval number for your protocol, approved by the IRB of Erasmus MC. 

2) Thank you for providing the underlying data and raw blot images as supplementary files (and for including legends for these within the files). Will you please include the legends for each supplemental data file in your' 'Supplementary Figure Legends' file? For example, the legend might read 'S1_data. Numerical data underlying Figure 1.'

PRESS

Thank you again for supporting Open Access publishing. We look forward to publishing your paper in PLOS Biology. 

Sincerely, 

Lucas Smith, Ph.D. 

Associate Editor 

PLOS Biology